# Cryogenic cave carbonates in the Dolomites (Northern Italy): insights into Younger Dryas cooling and seasonal precipitation

Gabriella Koltai[1], Christoph Spötl[1], Alexander H. Jarosch[2], Hai Cheng [3, 4, 5]

[1] Institute of Geology, University of Innsbruck, Innrain 52d, 6020 Innsbruck, Austria
[2] ThetaFrame Solutions, Hörfarterstrasse 14, 6330 Kufstein, Austria
[3] Institute of Global Environmental Change, Xi'an Jiaotong University, Xi'an, China
[4] State Key Laboratory of Loess and Quaternary Geology, Institute of Earth Environment, Chinese Academy of Sciences, Xi'an, China
[5] Department of Earth Sciences, University of Minnesota, Minneapolis, MN, USA

*Correspondence to*: Gabriella Koltai (gabriella.koltai@uibk.ac.at)

**Abstract.** In the European Alps, the Younger Dryas (YD) was characterized by the last major glacier advance with equilibrium line altitudes being ~220 to 290 m lower than during the Little Ice Age and also by the development of rock glaciers. Dating of these geomorphic features, however, is associated with substantial uncertainties leading to considerable ambiguities on the internal structure of this stadial, the most intensively studied one of the last glacial period. Here we provide robust physical evidence based on precise $^{230}$Th-dated cryogenic cave carbonates (CCC) coupled with thermal modelling indicating that early YD winters were only moderately cold in the Southern Alps. Our data argue for a negative temperature anomaly of $\leq 3°C$ in mean annual air temperature at the Allerød-YD transition in a mountain cave (Cioccherloch, 2274 m a.s.l.) in the Dolomites of northern Italy. Our data suggest that autumns and early winters in the early part of the YD were relatively snow-rich, resulting in a stable winter snow cover. The latter insulated the shallow subsurface in winter and allowed the cave interior to remain close to the freezing point (0°C) year-round, promoting CCC formation. The main phase of CCCs precipitation at ~12.2 ka BP coincides with the mid-YD transition recorded in other archives across Europe. Based on thermal modelling we propose that CCC formation at ~12.2 ka BP was most likely associated with a slight warming of approximately +1°C in conjunction with drier autumns and early winters in the second half of the YD. These changes triggered CCC formation in this alpine cave as well as ice glacier retreat and rock glacier expansion in the Alps.

## 1 Introduction

The last glacial period in the Northern Hemisphere (from 119 to 11.7 thousand years (ka) BP; Rasmussen et al., 2014) was characterized by abrupt climate shifts from cold and commonly arid stadials to mild and more humid interstadials. The youngest of these stadials is known as the Younger Dryas (YD or Greenland Stadial (GS) 1, ~12.8 to 11.7 ka BP; Rasmussen et al., 2014) and was a period when Northern Hemisphere temperatures returned to near-glacial levels, interrupting the last Termination.

The YD is among the most extensively studied periods in the late Quaternary due to the availability of high-resolution palaeoclimate records such as ice, marine and lacustrine sediment cores. Still, the forcing mechanism(s) for this cold episode remain debated (Alley, 2000; Baldini et al., 2018; Brauer et al., 2008; Broecker et al., 2010; Isarin et al., 1998; Renssen et al., 2015). The most widely accepted model invokes a near-shutdown of the Atlantic Meridional Overturning Circulation (AMOC) as a result of a major meltwater injection into the North Atlantic Ocean (e.g. Broecker et al., 1989), and the concomitant large-scale reorganization of the westerlies due to extensive winter sea ice formation (e.g. Bakke et al., 2009; Brauer et al., 2008). In a recent study using high-resolution speleothem records from Europe, as well-as Asian Monsoon- and South American Monsoon regions Cheng et al. (2020) demonstrated a north to south climate signal propagation via both atmospheric and oceanic processes at the onset of the YD. The authors also suggested that unlike other DO events, the termination of the YD was most probably initiated by a change in the western tropical Pacific and the Southern Hemisphere resulting in a strengthening of the AMOC.

The slowdown of the AMOC at the onset of the YD and the resulting southward displacement of the polar front and the westerlies led to cold conditions in N and NW Europe during prolonged winters (e.g. Broecker, 2006; Denton et al., 2005). In a recent study using a European-wide compilation of plant indicator species Schenk et al. (2018) suggested that YD summers remained relatively warm despite the AMOC slowdown with temperature decreases of 4.3ºC in NW Europe and 0.3ºC in E Europe relative to the preceding Bølling-Allerød interstadial (Greenland Interstadial (GI) 1). Using climate model simulations, Schenk et al. (2018) attributed relatively warm summers to atmospheric blocking induced by the Fennoscandian Ice Sheet preventing the penetration of cold westerly air masses entering Europe during the short summers. In contrast, blocking was almost absent during YD winters (Renssen et al., 1996). Overall, proxy data suggest that the YD climate was dominated by high seasonality and continentality across Europe with large meridional summer temperature gradients (e.g. Heiri et al., 2014a). Changes in winter climate were likely disproportionally larger (e.g. Broecker, 2006), but quantitative understanding of the YD climate remains heavily biased towards the summer given the scarcity of winter proxy data.

Speleothem-based palaeotemperature reconstructions from the Jura Mountains in northern Switzerland suggest a large drop of the mean annual air temperature (MAAT) of up to -10°C during the YD (Affolter et al., 2019; Ghadiri et al., 2018), while preliminary results of a similar study from a cave in western Austria suggest a much smaller difference (5.5°C) (Luetscher et al., 2016). Rock glacier records from the SE Swiss Alps argue for an even smaller cooling of only up to 3-4°C (Frauenfelder et al., 2001).

More recently, evidence for a time-transgressive climate shift mid-way through the YD has been reported from lacustrine sediments (e.g. Bakke et al., 2009; Brauer et al., 2008; Schlolaut et al., 2017), speleothems (Baldini et al., 2015; Bartolomé et al., 2015; Rossi et al., 2018) and marine sediments in Europe (Naughton et al., 2019). This climate shift has been attributed to a gradual northward movement of the polar front driven by the resumption of the AMOC and concomitant sea-ice retreat in the North Atlantic. The earliest indication of a climate shift during the mid-YD is recorded by a stalagmite from the Pyrenees, showing a gradual transition from dry to wet conditions starting at 12.45 ka BP (Bartolomé et al., 2015). It took about three hundred years for this shift to propagate to central and finally to northern Europe, where it is documented as a rapid change in

Ti content and varve thickness (Bakke et al., 2009; Lane et al., 2013). While many records from SW (Baldini et al., 2015; Bartolomé et al., 2015; Naughton et al., 2019; Rossi et al., 2018) and N Europe indicate that the first half of the YD was colder and drier than the second one, biomarker data from lacustrine sediments of the Gemündener Maar in W Germany suggest the opposite trend (Hepp et al., 2019).

In the Alps, climate information about the YD has been traditionally derived from studies of lake sediments (e.g. Grafenstein
et al., 1999; Heiri et al., 2014; Lauterbach et al., 2011) and glacier reconstructions (e.g. Ivy-Ochs et al., 2009; Kerschner et al., 2000; Kerschner and Ivy-Ochs, 2008; Moran et al., 2016), complemented by a few cave records (Li et al., 2020; Luetscher et al., 2016; Wurth et al., 2004). Studies on the internal structure of the YD, however, are rare and compromised by poor dating resolution, limiting our understanding of the mechanism(s) of the proposed mid-YD climate shift. Alpine paleoglacier records suggest an early YD glacier maximum between about 13.5 and 12.0 ka BP attributed to a combination of low temperatures
and enhanced precipitation differences between the northern, central and southern part of the Alps (Ivy-Ochs, 2015; Kerschner et al., 2016; Kerschner and Ivy-Ochs, 2008). Equilibrium line altitude reconstructions show that the inner zone of the Alps received ~20-30% less precipitation than today, mostly due to a decrease in winter precipitation, while annual precipitation in the Southern Alps was probably similar to modern values (Kerschner et al., 2016). Paleoglaciers from the Southern Alps show evidence of a double response, whereby the outermost and innermost moraines stabilized at ~12.3±0.7 ka and before 11.2±0.8
80    ka BP, respectively (Baroni et al., 2017; Ivy-Ochs et al., 2009).

In this study we focus on the Dolomites in the Southern Alps and provide seasonally resolved insights into the climate of the YD from cave deposits anchored by a precise [230]Th chronology. Given that the polar front reached as far south as the Alps during the early YD (Lane et al., 2013), this site is sensitive to track the northward migration of the polar front during the later YD. Rather than examining stalagmites commonly used in speleothem-based palaeoclimate research, we utilize a rather novel
speleothem variety which provides a uniquely robust temperature control: coarsely crystalline cryogenic cave carbonate (CCC for short). The leading genetic model envisages CCC formation under degrading permafrost conditions (e.g., Žák et al., 2018). Water ingresses into the cave when the seasonally thawing active layer of permafrost intersects the ceiling of the cave chamber, while most of the chamber is still within the permafrost, resulting in cave ice formation. Further climate warming leads to progressive degradation of permafrost and the cave air temperature slowly rises to 0°C. Drip water creates meltwater pools in
the cave ice bodies that freeze slowly triggering the precipitation of CCC. Regardless of the details of this model, the key point is that CCC form within perennial cave ice at temperatures very close to 0°C (Žák et al., 2018).

Using a  high-alpine cave whose paleothermal regime is assessed by heat-flow modelling, we use CCC data to argue against strong winter cooling during the early YD in the Southern Alps. Furthermore we demostrate that CCC formation in the shallow subsurface provides evidence of a maximum of 1-2°C annual warming at the mid-YD transition, and suggests that autumns
and winters may have become slightly drier in the second half of the YD compared to the early YD.

## 2 Study site

Cioccherloch is a single-entrance cave opening at 2245 m a.s.l. on the karst plateau of the Sennes region in the Dolomites (Fig. 1). The cave has approximately 250 m of passages. The entrance shaft is 20 m deep and intersects a subhorizontal cave level. A firn and ice cone is present at the bottom of this shaft fed by winter snow sliding down the shaft. Separated by a narrow squeeze which was excavated by cavers, a separate branch of the cave 60 m long and up to 10 m high descends from this upper cave level to approximately 55 m below the surface. Near the lower end of this gallery CCC were discovered for the first time in the Dolomites (Fig. 1). The air temperature at the CCC site monitored over a 1-year-period averages 2.5°C with minimum values (2.2°C) in January to March and maximum values (2.7°C) in November. These data show that the cave chamber is in thermal equilibrium with the outside MAAT at this elevation, obtained from nearby weather stations at Rossalm (2340 m a.s.l.) and Piz la Ila (2050 m a.s.l.) located less than 10 km from the study site. The mean air temperature is 2.2°C and 3.3°C at Rossalm and Piz la Ila, respectively (2015-2018; data source: Hydrographisches Amt, Autonome Provinz Bozen – Südtirol). The majority of snowfall in the Dolomites occurs from January to April with average snow heights of 4.2 and 3.4 m at Rossalm (2012-2019) and Piz la Ila (1999-2014), respectively. Autumn to early winter (September to December) snowfall amounts to an average of 1.0 and 0.8 m at the two weather stations.

## 3 Methods

### 3.1 Field work

CCC occurrences were mapped and samples were collected from five distinct heaps labelled A to E (Suppl. Fig. 1). In addition, small in-situ stalagmites were taken from the same chamber. Cave air temperature was recorded on an hourly basis using a Hobo Temp Pro v2 logger (Onset) between August 2016 and July 2017.

### 3.2 Morphological characterization

CCC samples were cleaned in an ultrasonic bath prior to documentation and laboratory analyses. Individual morphologies were examined using a Keyence VHX-6000 digital microscope.

### 3.3 Stable isotope analyses

CCCs samples were analyzed for their stable oxygen and carbon isotope composition using isotope ratio mass spectrometry (Spötl and Vennemann, 2003). In addition, two larger CCC particles were cut in half and micromilled at 0.1 to 0.3 mm resolution. The results are reported relative to the VPDB standard with a long-term precision better than ±0.08‰ (1σ) for both $\delta^{13}C$ and $\delta^{18}O$.

### 3.4 $^{230}$Th dating

Nineteen individual CCC particles were selected for $^{230}$Th dating. 15-20 mg of calcite was drilled using a handheld drill from the center of 17 crystals in a laminar flow hood. Two skeletal CCC crystals were analyzed as a whole as they were too small for aliquots to be drilled from them. Growth layers of a stalagmite (Cioc1) collected next to the CCC spots were drilled at three discrete horizons (2, 24 and 50 mm from the top) and prepared for analyses.

Ages were determined by measuring U and Th isotope ratios on a multi-collector inductively coupled mass spectrometer after their chemical separation following Edwards et al. (1987) and Cheng et al. (2013). Analyses were performed at Xian Jiaotong University (China). 2σ uncertainties for U and Th isotopic measurements include corrections for blanks, multiplier dark noise, abundance sensitivity, and contents of the same nuclides in the spike solution. Decay constants for $^{230}$Th and $^{234}$U were reported by Cheng et al. (2013). Corrected $^{230}$Th ages assume an initial $^{230}$Th/$^{232}$Th atomic ratio of (4.4 ±2.2) ×10$^{-6}$ and $^{232}$Th/$^{238}$U value of 3.8 as the value for material at secular equilibrium with the bulk earth. Final ages are given in years BP (before 1950 AD).

### 3.5 Thermal modelling

Heat conduction from the surface to 70 m depths was modelled using a 1d heat-flow model (https://zenodo.org/record/3982221). This model considers conductive heat transfer only and solves the heat equation (Eq. 1) utilizing finite differences as space discretizations alongside a forward Euler time-stepping scheme, stabilized with a diffusion-type Courant–Friedrichs–Lewy condition. The heat equation (Eq. 1) is described as

$$\frac{\partial T}{\partial t} = \alpha \frac{\partial^2 T}{\partial z^2} \tag{1}$$

where α is the thermal diffusivity, T is temperature and z is the vertical coordinate. The model assumes a homogenous host rock with no internal heat generation. Thermal diffusivity (α) of the limestone was set to 1.2 x 10$^{-6}$ m$^2$/s (Hanley et al., 1978) to account for some air-filled porosity.

As the lower boundary condition for the model, the ground heat flux was set to 0, to account for the presence of the shaft in the entrance zone of the cave, allowing exchange with the ambient air in this upper part of the cave above the CCC-bearing gallery, suppressing the minor effect of the geothermal heat flux. Modeling results are shown as MAAT against depth below the surface.

### 3.5.1 Scenario 1 – Allerød interstadial

We performed a series of model runs simuating possible climate scenarios for the YD (Table 1). Palaeotemperature estimates are based on published regional annual and summer air temperature reconstructions. All experiments started with a 1000 year-long Allerød climate (scenario 1) and a 2°C drop in MAAT compared to present day (i.e., ΔMAAT$_{Modern-Allerød}$=-2°C), consistent with regional proxy data (e.g., Ilyashuk et al., 2009). As stalagmite Cioc1 was actively growing in the Bølling interstadial (see Results), we assume that no permafrost was present at the site at the start of the interstadial. Hence a surface ground temperature of 1°C was used.

### 3.5.1 Scenario 2 – Stadial conditions during the early YD

The model output of scenario 1 was used as the starting condition for all scenario 2 runs. Scenarios 2a to 2e were used to model the evolution of subsurface conditions during the early YD. As the mid-YD transition was determined at 12,240 ± 40 varve years BP at Meerfelder Maar, Germany (Lane et al., 2013), scenario 2 models were run for 610 years (i.e. from 12.85 to 12.24 ka BP).

As a first step we modeled the penetration of the seasonal signal without the presence of winter snow (scenarios 2a, 2b, 2d) to

provide an endmember for early YD cooling. Scenario 2a was forced with a large atmospheric cooling ($\Delta MAAT_{\text{Allerød-early YD}}$ = -7°C) in accordance with speleothem-based paleotemperatures from northern Switzerland (Affolter et al., 2019; Ghadiri et al., 2018). This experiment used MAAT of -6.5°C (i.e. -20°C in January; Table 1) to model the thermal evolution of the subsurface. In scenario 2b we applied a smaller cooling of $\Delta MAAT_{\text{Allerød-early YD}}$ = -3°C (i.e. MAAT of -2.5°C and -13°C in January; Table 2), as suggested by stalagmite fluid inclusion data from western Austria (Luetscher et al., 2016). Scenario 2d

considers an even smaller cooling ($\Delta MAAT_{\text{Allerød-early YD}}$ = -2.5°C), as suggested by rock glacier records from the Southern Alps (Frauenfelder et al., 2001) and a MAAT of -2°C was used (i.e. 12°C in January, Table 1).

In a second step we included the buffering effect of a stable winter snowpack insulating the ground from the winter chill. This buffering effect (expressed as snow $\Delta T$) was set to its maximum in scenarios 2c and 2e to test if a similar amplitude of cooling investigated in scenarios 2b and 2d (Table 1) would allow CCC formation in the presence of a winter snow cover. Experiment

2c used a MAAT of -2°C and simulated the maximum possible buffering of the winter snowpack of 5°C (i.e. snow $\Delta T$ of 5°C), resulting in mean annual effective temperature (MAET) at the base of the snowpack of -1.3°C (Table 1).

In scenario 2e the thermal buffering of the snow cover was set again to its maximum value and we used a snow $\Delta T$ of 4.7°C in combination with a MAAT of -2°C, resulting in a MAET of -0.9°C (Table 1). $\Delta T$ values of 5°C and 4.7°C are considered to be realistic for the YD at this alpine setting. Measurements in modern permafrost areas in NE China reported mean ground

surface temperatures to be up to 9°C higher with 10-25 cm snow cover during winter resulting in an annual net thermal effect (i.e. snow $\Delta T$) of 3 to 5°C of a 10 to 40 cm thick snow cover (Zhang et al., 2005). Similar values were reported from the permafrost regions of the Artic in Russia (Zhang et al., 2005)

### 3.5.3 Scenario 3 – Stadial conditions during the late YD

Three simulations investigate the influence of climate change at the mid-YD transition. Scenario 3a simulates a rapid reduction

in winter snow cover without a temperature change with respect to scenario 2e (Table 1). We kept MAAT constant at -2.0°C and reduced snow $\Delta T$ from 4.7°C to 2.0°C (Table 1).

In scenarios 3b and 3c we applied an instantaneous warming of +1°C in MAAT compared to scenario 2c. In these two runs summer temperatures were kept constant and a small amount of winter warming was allowed. We kept the climate warming at 1°C (i.e. $\Delta MAAT_{\text{Allerød-late YD}}$ = -2°C), because a larger warming would result in a climate similar to the preceding Allerød.

While scenario 3b only considers atmospheric warming, we allow some late summer and autumn snow accumulation in scenario 3c (snow $\Delta T= 2.5°C$, Table 1).

## 4 Results

### 4.1 CCC morphology

CCC occur as loose crystals and crystal aggregates in small heaps on and partly underneath five breakdown blocks (Suppl.
Fig. 1). Individual crystals and aggregates thereof come in a variety of shapes and sizes, the largest reaching 1.4 cm in length. Morphologies include amber-colored crystals and crystal aggregates of rhombic, raft, beak-like and split crystal habits (Suppl. Fig. 1). Split and beak-like crystals are most abundant (heaps A, B, and D). Translucent skeletal crystals are rarely present in heaps B and D, but are dominant in C and E.

### 4.2 Stable isotope composition

CCC show high $\delta^{13}C$ values varying from 1.3 to 5.4‰ and low $\delta^{18}O$ values ranging from -21.8 to -10.1‰ (Fig. 2, Suppl. Table 1). These values fall within the compositional range characteristic of CCC (Žák et al., 2018 and references therein), confirming their precipitation in slowly freezing of pockets of water enclosed in cave ice. A beak-like crystal, 4.5 mm in diameter, revealed ≤1.3‰ and ≤1.2‰ internal variability in $\delta^{13}C$ and $\delta^{18}O$, respectively, while a 5 mm large rhombohedral crystal shows ≤0.4‰ and ≤0.2‰, respectively. Holocene stalagmites from the same chamber exhibit distinctly different stable isotope values with
$\delta^{13}C$ and $\delta^{18}O$ values ranging from -7.6 to -2.0‰ and from -9.1 to -7.4‰, respectively (Fig. 2).

### 4.3 $^{230}Th$ dating

The $^{238}U$ concentration of CCC samples varies from 0.8 to 2.2 ppm (Table 2). Four samples yielded $^{230}Th/^{232}Th$ atomic ratios less than $100\times10^{-6}$, indicating significant detrital contamination (Table 2) and the corresponding ages were excluded. The 2σ precision of the remaining 13 ages ranges from 0.4 to 2.7%.

All CCC ages fall within the YD (as defined by Rasmussen et al., 2014), whereby individual ages show a spread from 12.61 ±0.22 to 12.08 ±0.19 ka BP. Although most of the ages overlap within their 2σ errors, some $^{230}Th$ ages suggest that CCC formation commenced at the beginning of the YD (i.e. 12.61 ±0.22 ka BP). On the other hand, the error-weighted mean of all ages yielded 12.19 ±0.06 ka BP and 90% percent of $^{230}Th$ dates cluster at ~12.2 ka BP. There is no systematic age difference between samples from individual heaps nor between different CCC morphologies.

In contrast to CCCs, the $^{238}U$ concentration of stalagmite Cioc1 is much lower (~0.5 ppm). $^{230}Th/^{232}Th$ atomic ratios vary between $34\times10^{-6}$ and $1570\times10^{-6}$ (Table 2). The resulting $^{230}Th$ ages demonstrate that stalagmite growth commenced during the Late Glacial at 14.98 ±0.14 ka BP. Petrographic analysis provides strong evidence for a growth interruption after which calcite deposition re-started in the mid-Holocene at 5.88 ±0.15 ka BP and continued until 1.32 ±0.27 BP ka.

## 4.4 Thermal modelling

We performed a series of model runs covering possible climate scenarios for the YD (Table 1) to explore the relationship between atmospheric temperature changes and the temperature 50 m below the surface at this high-elevation site. We define the thermal boundary conditions for CCC formation as -1 to 0°C ("CCC window" - see discussion) whereby the likelihood of CCC formation is highest between -0.5 and 0°C. Paleotemperature estimates used in these computations are based on regional annual and summer air temperature reconstructions.

### 4.4.1 Scenario 1 – Allerød interstadial


In this scenario we simulated an interstadial, similar to the 1000 year-long Allerød preceding the YD. This experiment shows that after 1000 years a temperature of 0.5°C is reached at 50 m depth (Suppl. Fig. 2).

### 4.4.2 Scenario 2 – Stadial conditions during the early YD

In the next five experiments (2a – 2e) we explored the timing of perennial cave ice development and tested whether water
pockets in cave ice could have experienced slow freezing during an early YD characterized by cold stadial conditions.
The results of scenario 2a show that the atmospheric cooling rapidly propagates into the subsurface resulting in the development of permafrost down to 50 m depth in less than 50 years (Fig. 3a, Suppl. Fig. 3a) in case of a -7°C cooling compared to the preceeding Allerød interstadial.

Modelling results (scenario 2b) show that the cave 50 m below the surface cools to -1°C in about 100 years in case of an
extremely dry YD characterized by a -3°C drop in MAAT compared to the Allerød interstadial (scenario 2b, Fig. 3b, Suppl. Fig. 3b). The presence of a snowpack (2c) delays this subsurface cooling and results in temperatures at the CCC site between 0.5 and -1°C within 150 years from the start of the atmospheric cooling (Fig. 3c, Suppl. Fig. 3c).

The last two early YD experiments (2d and 2e) consider even milder winters ($\Delta MAAT_{Allerød-early\ YD}$ = -2.5°C, Table 1). With no winter snow cover (2d) the temperature at the CCC site reaches -0.5°C after ~60 years and drops below -1°C (i.e. it leaves
the "CCC window") after 100 years (Fig. 3d, Suppl. Fig. 3d). On the other hand, if snow insolates the ground in winter and buffers the winter cold by 4.7 °C (i.e. $\Delta T$ of 4.7°C; Table 2), temperatures at 50 m depth stay above -0.8°C for an extended period of time (Fig. 3e, Suppl. Fig. 3e).

### 4.4.3 Scenario 3 – Stadial conditions during the late YD

In the first late YD experiment (3a) we examined how permafrost conditions would change with increasing aridity in autumn
and early winter compared to the early YD (2e), resulting in a reduction in winter snow cover. As the insulating effect of the winter snowpack is reduced, the depth zone of the CCC site experiences rapid cooling and approaches -1.8°C after 100 years, leading to permafrost development, inconsistent with stable conditions near 0°C required for CCC formation (Fig. 4a, Suppl. Fig. 4a).

The results of scenario 3b demonstrate that even though winters become slightly less cold, the subsurface at 50 m depth nevertheless cools from -1.3°C to -1.5°C due to the lack of a winter snow cover. This scenario is not compatible with CCC formation as it leads to permafrost aggradation (Fig. 4b, Suppl. Fig. 4b). In contrast, even the presence of a moderate snow cover (3c) would allow the subsurface at 50 m depth to slowly warm to -1°C after 75 years of the start of the atmospheric warming (Fig. 4c, Suppl. Fig. 4c), creating favorable conditions for slow freezing of liquid water pockets in the cave ice introduced by dripping water.

## 5. Discussion

### 5.1 Conditions close to 0°C in the shallow subsurface

CCCs form in slowly freezing water pockets enclosed in cave ice when the cave interior temperature is just slightly below 0°C (e.g., Žák et al., 2018). Although the deposition of CCCs in many cases mark climate transitions (e.g. Luetscher et al., 2013; Richter and Reichelmann, 2008; Spötl and Cheng, 2014; Žák et al., 2012), the large size of some CCCs (up to 50 mm in caves elsewhere, unpublished data from our group) and [230]Th ages from their central and rim areas (unpublished data from our group) argue for very stable cave microclimate conditions for at least several years.

CCCs (Table 2) provide unequivocal evidence that perennial ice was present in the lower descending gallery of Cioccherloch during the first part of the YD (Fig. 5). To demonstrate the likelehood of CCC formation, the Kernel density function of all [230]Th ages was calculated. As the majority of [230]Th ages overlap within their 2σ errors, it is not possible to determine whether CCC formation took place semi-continuously for 400-600 years or if they represent two different generations clustering at ~12.6 and ~12.2 ka BP (Fig. 5). Overall, CCCs in Cioccherloch record negative interior cave air temperatures very close to 0°C from ~12.6 to ~12.2 ka BP, initiating progressive freezing of meltwater pockets in perennial ice which were created by drip water.

The air temperature in the homothermic zone of caves reflects the MAAT of the outside atmosphere. Ice-bearing caves, however, commonly represent an exception to this rule (e.g., Luetscher and Jeannin, 2004; Perșoiu, 2018 and references therein). Due to its descending geometry lacking a lower entrance, the entrance shaft of Cioccherloch is a cold trap, as evidenced by the snow and ice cone at its base. The resulting negative thermal anomaly, however, is restricted to the upper cave level close to this snow cone. Although today the descending gallery with the CCC occurrences experiences a small intra-annual temperature variation (~0.5°C), the MAAT in this gallery matches the ambient MAAT (2.5°C). If the CCC-bearing chamber had been ventillated in the past the fine crystalline variety of CCCs would have formed by rather rapid freezing of water films (e.g. Žák et al., 2008) instead of coarse crystalline CCCs that require stable thermal conditions. Today this gallery is connected to the the cave´s upper level via a narrow squeeze but prior to cave exploration in the 1980s and 1990s, this squeeze was partly closed by rubble. Therefore, we presume that this descending gallery was essentially thermally decoupled from the upper cave level in the past and hence unaffected by the negative thermal anomaly. As a result, CCCs in the lower gallery record changes in the thermal state of the subsurface as a consequence of atmospheric temperature changes. Given the

lack of ventilated shafts connecting this gallery to the surface, we argue that heat exchange between the latter and the gallery occurred primarily via conduction. Additional heat transfer occurred via drip water and minor air advection via small-scale fissures in the ceiling cannot be ruled out. The slow flowing seepage water leading to meltwater pockets in the cave ice, however, is likely thermally equilibrated with the ca. 50 m-thick rock above the cave and given its very low discharge carries

comparably little heat from the surface. Advective processes are difficult to quantify for any time in the past and no attempts were made to include them in the thermal model. Qualitatively, both processes would increase the rate of temperature change in this gallery as a response to atmospheric change above the cave.. By considering heat conduction only, our simulations yield quantitative constraints on the maximum duration of temperature changes propagated into the shallow subsurface.

The formation of CCCs near the lower end of the descending gallery requires the 0°C isotherm to be located about50 m below

the ground surface atthis high-elevation site in the Dolomites during the YD. On the other hand, stalagmite Cioc1 provides strong evidence that the air temperature in this gallerry was constantly above 0°C during the Bølling-Allerød interstadial. $\delta^{13}$C values of Cioc1 show a decrease from -2.5 to -6.6‰, suggesting gradual soil development above the cave (unpublished data by the authors). During the early YD, atmospheric cooling led to the aggradation of cave ice. During the mildYD summers (from ~12.6 to ~12.2 ka BP) drip water sourced from snowmelt and/or rain created meltwater pools on the ice that subsequently

underwent slow freezing at cave air temperatures ~2-3°C (i.e. 2.5 ± 0.5°C) lower than today.

CCC deposition in caves has traditionally been attributed to near-surface permafrost degradation in response to atmospheric warming (e.g. Žák et al., 2018 and references therein). In this case however, stalagmite growth (Cioc1) in Cioccerloch cave indicates cave air temperatures above 0°C and the presence of dripping water in the cave at the Bølling-Allerød interstadial. Thus, CCC formation starting at 12.6 ka cannot represent a delayed response to the Bølling-Allerød interstadial warming.

Instead our data suggest that CCCs in Cioccherloch formed during a cold climate associated with permafrost aggradation. Therefore, we hypothesize that analogous to climate warmings possible "CCC windows" opened during the transition into stadials and remained open for a variably long period of time depending on the local thermal conditions of the subsurface.

## 5.2 Magnitude of YD cooling

Our thermal model provides quantitative constraints on how the cave air temperature evolved in response to different climate

conditions. The model uses two input parameters: MAAT and the insulating effect of winter snowpack. A recent study by Schenk et al. (2018) suggests that YD summers remained relatively warm with a temperature decreases of 4.3°C in NW Europe and 0.3°C in E Europe relative to the preceding Bølling interstadial. We therefore kept the July temperatures 3°C to 4°C lower than modern values (Table 2) and attributed most of the MAAT change to winter cooling. As the buffering effect of the snow is set to its maximum value to counteract the winter chill, this approach allows the reconstruction of the possible maximum

amplitude of MAAT (winter) cooling of the surface.

CCCs dated to the first and second half of the YD indicate conditions very close to 0°C for an extended period of time during this stadial. Our thermal model shows that CCC formation starting at 12.6±0.2 ka BP at this sensitive mountain site recquires a moderate atmospheric cooling at the Allerød-YD transition of -4.5 to -5°C relative to today (Fig. 5). Without a winter snow

cover (scenarios 2b and 2d) the "CCC window" would open too early and close quickly afterwards, inconsistent with the CCC ages (Fig. 5). Scenarios 2c and 2e show that if the YD climate was characterized by a -5 to -4.5°C drop in MAAT relative today (i.e. $\Delta MAAT_{Allerød-YD}$= -3 to -2.5°C), a stable snow cover during winter is needed to prevent the cave from freezing, effectively shielding the ground from the cold stadial winters (cf. Zhang, 2005). Scenario 2e including a winter snow cover provides the best fit with the CCC data giving rise to a 400 year-long period characterized by a very slow cooling of the subsurface with cave air temperatures near -0.8°C.

Our data do not support the notion of a very cold YD in the Alps($\Delta MAAT_{Allerød-YD}$ = -7 to -8°C; scenario 2a) as suggested by speleothem data from low-lying caves in northern Switzerland (Affolter et al., 2019; Ghadiri et al., 2018). Such a drastic lowering of the MAAT would freeze Cioccherloch rapidly even if a thin winter snowpack was present, and would result in rather abrupt development and deepening of permafrost, preventing CCC formation (Fig. 7). Such a stark cooling would in fact lead to climate conditions similar to the Last Glacial Maximum (LGM), for which noble gas data from groundwater studies around the Alps suggest 7-10°C lower temperature compared to the Holocene (e.g., Šafanda and Rajver, 2001; Stute and Deak, 1989; Varsányi et al., 2011) and which would inevitably lead to the build-up of glaciers at this elevation in the Dolomites. Our data, however, are consistent with observations from rock glaciers (Frauenfelder et al., 2001) and lake sediments in the Swiss Alps (Von Grafenstein et al., 2000) suggesting a moderate cooling at the Allerød-YD transition. A fluid inclusion-based paleotemperature reconstruction using stalagmites from Bärenhöhle in western Austria also indicates a maximum temperature drop of about 5.5°C, supporting our interpretation (Luetscher et al., 2016).

### 5.3 Increased seasonality in the early YD

CCCs provide a uniquely robust control on cave air temperatures and consequently on the MAAT above the cave. Our data argue for a ≤3°C drop in MAAT at the Allerød-YD transition, but provide no direct information on the seasonal cycle of ambient atmospheric temperatures. A recent multi-proxy-model comparison by Schenk et al. (2018) suggests persistently warm summers during the YD with a median regional cooling of 3°C and 0.3°C over NW- and E-Europe, respectively, compared to Bølling-Allerød summers. These authors also argue that previous studies using chironomids overestimated the YD cooling signal. A similar amplitude of change is suggested for July air temperatures by lake records from the Western Alps. Pollen and cladocera-inferred temperature reconstructions indicate a summer cooling of 2-4°C at the Allerød-YD transition at Gerzensee (Swiss Plateau, e.g., Lotter et al., 2000), consistent with a 3.5°C drop in July air temperatures reported from Maloja Pass (Fig. 6) in eastern Switzerland (Ilyashuk et al., 2009). We therefore consider 0.3° and 4°C as minimum and maximum estimates of YD summer cooling, respectively, relative to the Bølling-Allerød. As our CCC data in conjunction with thermal modelling constrain the drop in MAAT at the Allerød-YD-transition to ≤3°C, they allow to calculate the maximum possible increase in seasonality at the Allerød-YD transition. If we assume that YD summers were only 0.3°C colder than in the Allerød by using the lowest summer temperature change proposed by Schenk et al. (2018), we find that early YD winters at 2270 m a.s.l. were no colder than -13.7°C (mean January temperature). This argues for an enhanced seasonality in the Dolomites, whereby the winter-summer temperature difference increased by up to 5.4°C at the Allerød-YD transition (i.e. ≤ 22.4°C in the YD).

Thermal modelling shows that a thin winter snowpack effectively shielding the subsurface during the cold winters is needed to account for CCC formation commencing at 12.6 ±0.2 ka at Cioccherloch. Studies in modern permafrost areas suggest that a stable winter snow cover of only ~35 cm results in a positive shift (i.e. snow $\Delta T$) of up to 5.5°C in the mean annual ground surface temperature (Zhang, 2005). Changes in the timing, duration, thickness and density of the snow cover may promote either the development or the degradation of permafrost (Zhang, 2005). While a snowpack in the cold season leads to a positive ground temperature anomaly, a summer snow cover insulates the ground from warm air and facilitates the development of permafrost. In a study of Arctic permafrost, Park et al. (2014) found that the thermal state of the underlying soil is more affected by early winter than peak winter snowfall. Therefore, we argue for a moderately humid early YD with snowfall during fall and early winter.

Proxy data show a heterogeneous picture with repect to precipitation patterns in the Alps during the YD (Belli et al., 2017; Kerschner et al., 2016; Kerschner and Ivy-Ochs, 2008). The distribution of rock glaciers in the eastern Swiss Alps suggests a 30-40% reduction in precipitation compared to today (Frauenfelder et al., 2001). Paleoglacier records for the Central Alps point to a similar reduction, mostly due to a reduction in winter precipitation. The few paleoglacier records in Southern Alps suggest that precipitation sums were similar to modern day values but winter precipitation was probably reduced compared to modern day leading to enhanced seasonal contrasts (Kerschner et al., 2016). A recent study compiling data from 122 paleoglaciers, shows that the southerly displacement of the the polar frontal jet stream led to cold air outbreaks and increased cyclogenesis over the Mediterranean and negative precipitation anomalies over the Alps compared to modern day (Rea et al., 2020). Precipitation pattern during the early YD closely resembled modern Scandinavian circulation patterns, resulting in an autumn to early winter (September to November) and mid-winter (december to February) precipitation increase over the Mediterranean (Rea et al., 2020). This is consistent with our findings, which argue that early YD autumns and early winters remained relatively snow-rich in the Southern Alps, although the amount of annual precipitation may have been reduced.

### 5.4 Climate change during the mid-YD

High-resolution speleothem (Baldini et al., 2015; Bartolomé et al., 2015; Rossi et al., 2018) and lake records (Bakke et al., 2009; Brauer et al., 2008; Lane et al., 2013) from W and N Europe suggest a time-transgressive change in atmospheric circulation during the YD associated with a warming of parts of Europe due to a retreat of winter sea ice and a northward migration of the polar front (e.g., Baldini et al., 2015; Bartolomé et al., 2015). At Meerfelder Maar, Germany, this so-called mid-YD transition occurred at 12.24 ±0.04 ka BP (Lane et al., 2013). This timing is strikingly similar to the error-weighted mean of the CCC dates from Cioccherloch (12.19 ±0.06 ka BP), suggesting that the main phase of CCC formation at this site may have been related to climate change. Our thermal simulations provide important constraints on the type and magnitude of climate change during the mid-YD transition at this high-alpine site and suggest a mild warming by up to 1°C (MAAT) with a slight reduction in precipitation.

A change from moderately snow-rich to snow-poor autumns and early winters from the early to the late YD combined with a small atmospheric warming (scenario 3c) is consistent with the main advance of Alpine ice glaciers (Fig. 6) during the first

few centuries of the YD (Baroni et al., 2017; Heiri et al., 2014b; Ivy-Ochs et al., 2009). The subsequent glacier reduction and the parallel increase in rock-glacier activity advocate less humid conditions towards the end of the YD and the early Holocene in the Western and Eastern Alps (Ivy-Ochs et al., 2009; Kerschner and Ivy-Ochs, 2008; Sailer and Kerschner, 1999). Speleothem records from the Pyrenees (Baldini et al., 2015; Bartolomé et al., 2015), Cantabrian Cordiella (Rossi et al., 2018) and the Adriatic coast (Belli et al., 2017) suggest a precipitation increase in the late YD, attributed to a strengthening of the westerlies. Precipitation-sensitive archives in the northern Alps, however, do not show evidence of such a change in the hydroclimate. Benthic ostracod $\delta^{18}$O records from Lake Ammersee (Grafenstein et al., 1999) and Lake Mondsee (Lauterbach et al., 2011) only show a gradual ca. 1‰ increase across the YD, arguing against a major step-wise change in the precipitation regime as would be expected as a result of the migration of the polar front across the Alps (Fig. 6). This slight increase in $\delta^{18}$O values, however, is compatible with our interpretation of a small decrease in autumn precipitation in the northern alpine catchment areas of these lakes in the second half of the YD (Fig. 6) coupled with a minor ($\leq 1$°C) warming. Speleothem $\delta^{18}$O data from the Jura Mountains (Affolter et al., 2019) and from Hölloch Cave west of the Ammersee catchment (Li et al., 2020; Wurth et al., 2004) likewise lack isotopic evidence of a significant climate change within the YD. A recently published replicated high-resolution $\delta^{18}$O speleothem record from Hölloch Cave rather exhibits a gradual increase in $\delta^{18}$O across the YD (Li et al., 2020) similar to northern Alpine lake records. While our data and paleoglacier evidence are consistent with a slight warming at the mid-YD transition, they both argue for a reduction in fall and winter precipitation. This suggests that the popular model of a south-north migration of the polar front and a concomitant increase in westerly-driven precipitation (e.g., Lane et al., 2013) is too simplistic at least for the greater Alpine realm, underscoring the need for regionally resolved paleoclimate models. In fact, even at the key site of Meerfelder Maar, the proposed increase in (winter) precipitation is poorly captured by most proxy data except for the abundance of Ti, which is attributed to spring snow melt (Lane et al., 2013).

## 6 Conclusions

This article presents the first record of CCC in the Dolomites which in contrast to many studies from Central European caves, formed not during a major climate warming but within a prominent stadial. These deposits indicate sustained negative temperatures close to ~0°C between ~12.6 and ~12.2 ka BP at about 50 m below the surface, initiating the slow freezing of dripwater-induced meltwater pockets in perennial cave ice. Combined with a thermal model the high-elevation setting of this cave suggests a $\leq 5$°C drop in MAAT compared to today, incompatible with extreme winter cooling during the YD. CCC formation during the early YD requires autumn to early winter snowfall forming a sufficiently thick and stable snow cover insulating the ground from the winter cold. CCC formation during the early YD coincided with the maximum YD extent of Alpine glaciers, consistent with abundant snowfall in autumn and winter and with decreased summer temperatures. Using a 0.3-4°C cooling for the short and mild early YD summers as suggested by data-model comparison studies (Heiri et al., 2014b;

Schenk et al., 2018), mean January air temperatures at this alpine site were most likely not colder than -13.7°C. Seasonal temperature differences between early YD summers and winters were therefore up to 5.7°C larger than during the Allerød.

The [230]Th data provide strong evidence that CCC formation at ~12.2 ka occurred in response to climate change associated with the mid-YD transition. CCC formation at this high-alpine cave advocates for a small atmospheric warming (i.e. +1°C in MAAT) and a reduction in fall precipitation in the late YD. We propose a shift from snow-rich early YD towards snow-poor late YD autumns and early winters, which is consistent with the retreat of YD glaciers in the Alps and an increase in rock glacier activity.

CCCs are a novel paleoclimate archive allowing to precisely constrain permafrost thawing events in the past. Our study demonstrates that CCCs can also provide quantitative constraints on paleotemperature and seasonally resolved precipitation changes.

**Code and data availability**

The code for the 1d heat-flow model is available online (https://zenodo.org/record/3982221). Data is included in Tables 1 and 2 and additionally given in Supplementary Table 1.

**Acknowledgements**

Field work and sampling was carried out in cooperation with the Naturpark Fanes-Sennes-Prags under a permit issued by the Amt für Naturparke, Abt. 28. Natur, Landschaft und Raumentwicklung of the Autonome Provinz Bozen – Südtirol. Gottfried Nagler, Andreas Treyer and Charlotte Honiat provided support during fieldwork and Jia Xue measured two [230]Th ages. The Hydrographische Amt of the Autonome Provinz Bozen – Südtirol kindly provided meteorological data. Paolo Mietto is acknowledged for sharing information on the caves explored by the speleo club Proteo, Jeffrey S. Munroe, Yuri Dublyansky and Hanns Kerschner and Paul Töchterle for fruitful discussions that helped to improve the manuscript. Paul Töchterle calculated Kernel density functions. Stéphane Affolter is thanked for providing data from Milandre Cave. This work was supported by the Autonome Provinz Bozen – Südtirol, Amt für Wissenschaft und Forschung (grant 3/34 to C.S.) and the Tiroler Wissenschaftsförderung grant WF-F.16947/5-2019 (to G.K.).

**Author contribution**

G.K. and C.S. designed the study, carried out field work, performed petrographic, stable isotope analyses and heat-flow modelling. G.K. carried out uranium-series dating, supervised by H.C. A.H.J developed the heat flow model. G.K. wrote the paper with contributions from all co-authors.

**Competing interests**

The authors declare that they have no conflict of interest.

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

## Figures

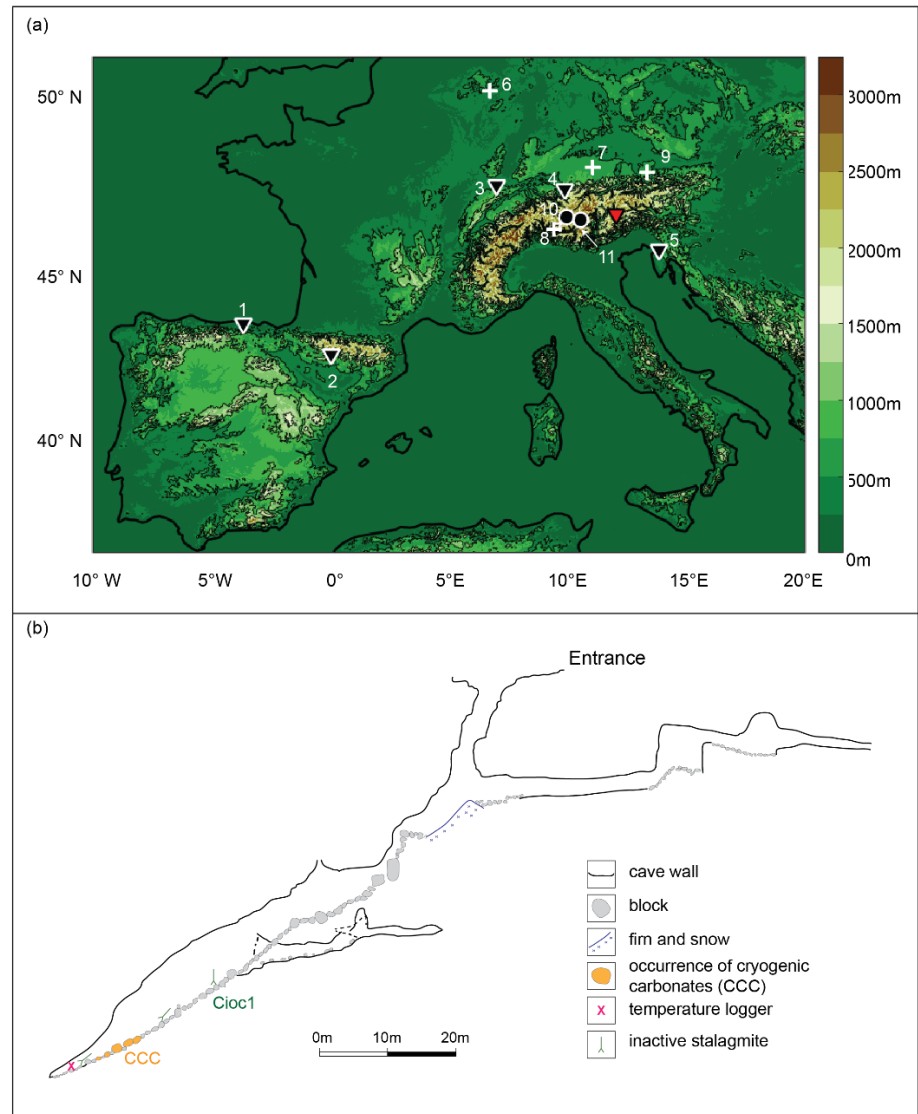


**Figure 1: Relief map of Europe showing the location of Cioccherloch (red triangle) and other European YD speleothem (black triangle), lacustrine (white cross) and palaeoglacier (black circle) records (a) mentioned in the text (1 −La Garma cave, 2 − Seso cave, 3 − Milandre Cave, 4 −Bärenhöhle, 5 −Savi Cave, 6 −Meerfelder Maar, 7 − Lake Ammmersee, 8 −Majola Pass, 9 − Mondsee 10 −Err-Julier and Julier Pass, 11 −Ortles). Vertical cross section of Cioccherloch Cave showing the CCC findings in the terminal chamber (b).**

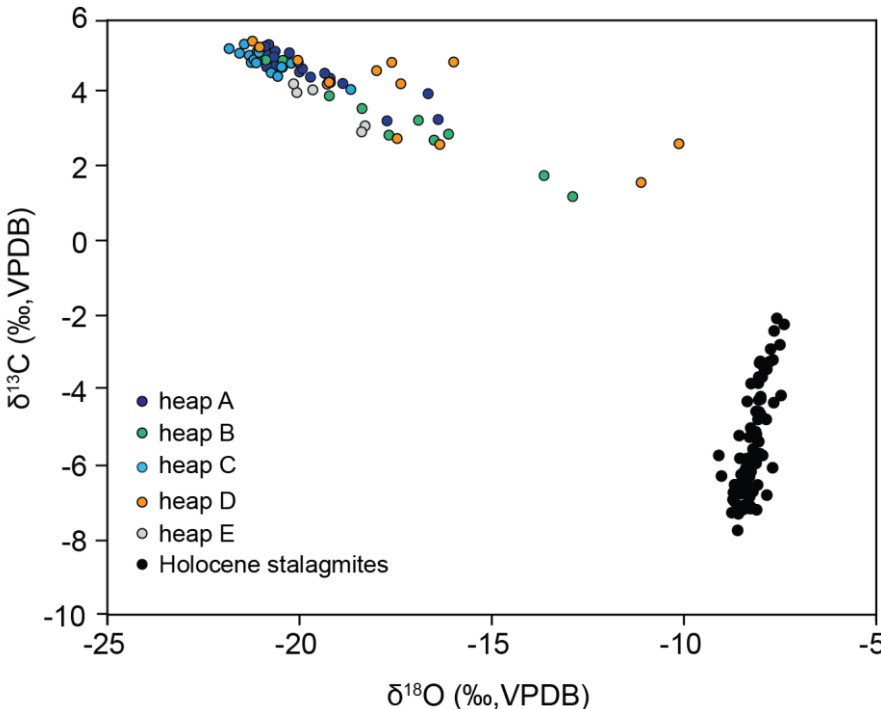

**Figure 2: Stable isotope composition of CCCs from Cioccherloch Cave. Samples from different CCC heaps are color-coded. Values of two Holocene stalagmites from the same cave gallery are shown for comparison (unpublished data by the authors).**


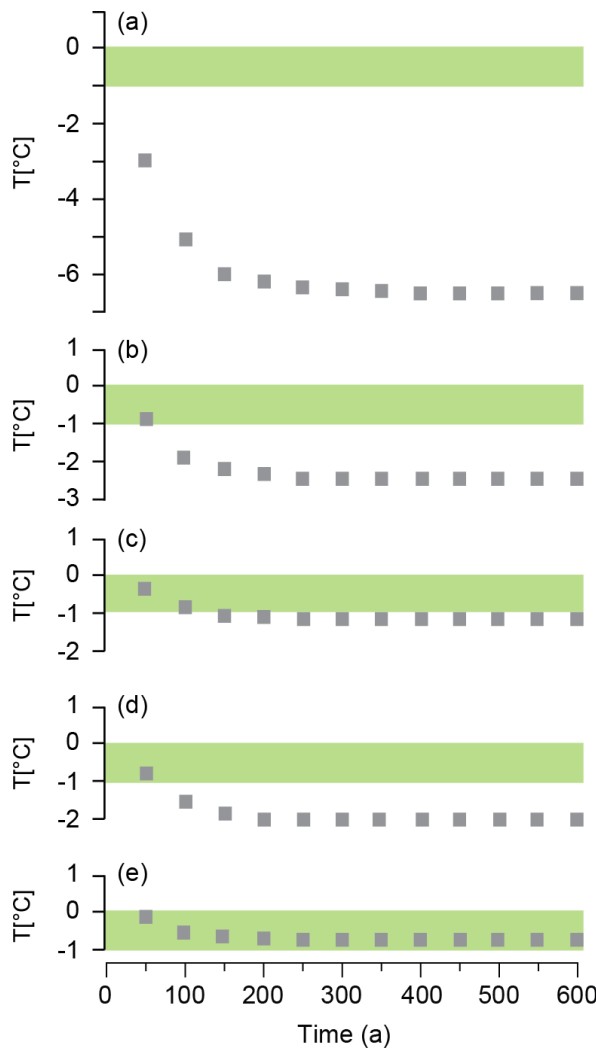

**Figure 3: Thermal models 2a to 2e simulating the development and deepening of the permafrost during the early YD. Modeling results depict the ground temperature at 50 m depth at the depth of the CCC site in Cioccherloch. These models use the temperature profile of scenario 1 as initial condition. Model 2a (a) is forced with a MAAT of -6.5°C. Models 2b (b) and 2c (c) use a MAAT of -2.5°C and the latter simulates the impact of a winter snow cover resulting in the attenuation of the winter cold by 5°C (i.e. snow ΔT = 5°C). Scenarios 2d (d) and 2e (e) model the changes of the thermal profile at the CCC site using a MAAT of -1.5°C. Scenario 2e considers the presence of a winter snow pack resulting in snow ΔT of 4.7°C. The green horizontal bar marks the -1 to 0°C "window" of possible CCC formation for the depth range of the CCC site in this cave.**


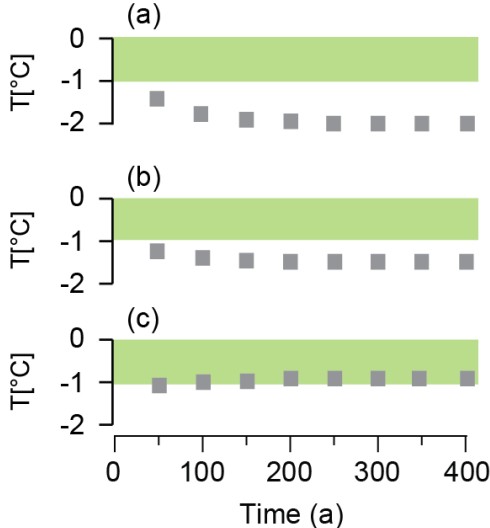


**Figure 4: Thermal model runs simulating a change in YD climate starting at 12.24 ka (mid-YD transition). Results show the ground temperature at 50 m depth (grey rectangles), at the depth of the CCC site in Cioccherloch. Model 3a (a) simulates the subsurface thermal conditions for a MAAT identical to scenario 2e (i.e. MAAT -2.0°C) but the autumns and winters became drier resulting in a reduction of the winter snow cover (snow ΔT=2.0°C). Scenarios 3b (b) and 3c (c) simulate the impact of a +1°C rise in MAAT with**

**respect to scenario 2c (i.e. MAAT -1.5°C) and the 3c model also includes a thin winter snow cover (snow ΔT=2.5°C). The green horizontal bar marks the -1 to 0 °C "window" of possible CCC formation at the depth of the CCC site in Cioccherloch.**

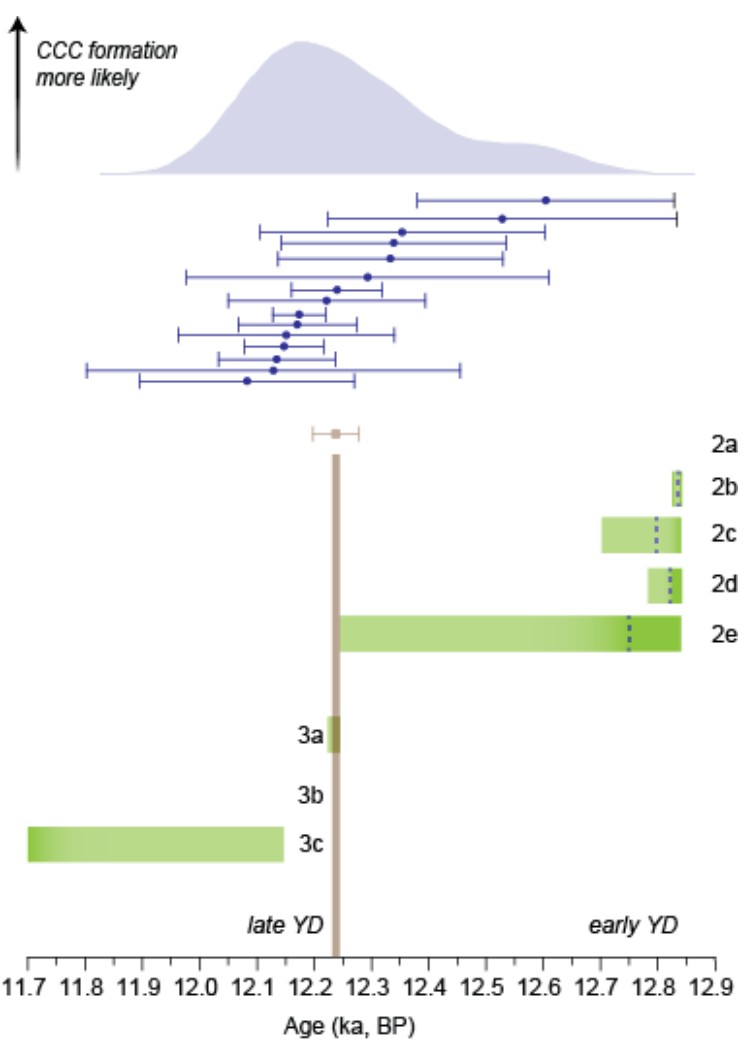

Figure 5: [230]Th ages of CCC and their 2σ uncertainties from Cioccherloch (blue bars) and CCC formation "windows" as suggested by model scenarios for the early and late YD. Calculated Kernel density function of the [230]Th ages shows the likelihood of CCC formation (blue shaded area). The green bars mark the -1 to 0°C window of possible CCC formation at the depth of the CCC site. The blue dashed vertical lines mark the -0.5°C isotherm at 50 m depth (2c-2e). The brown vertical line and age with the 2σ error bar mark the timing of the mid-YD transition at Meerfelder Maar (Lane et al., 2013).

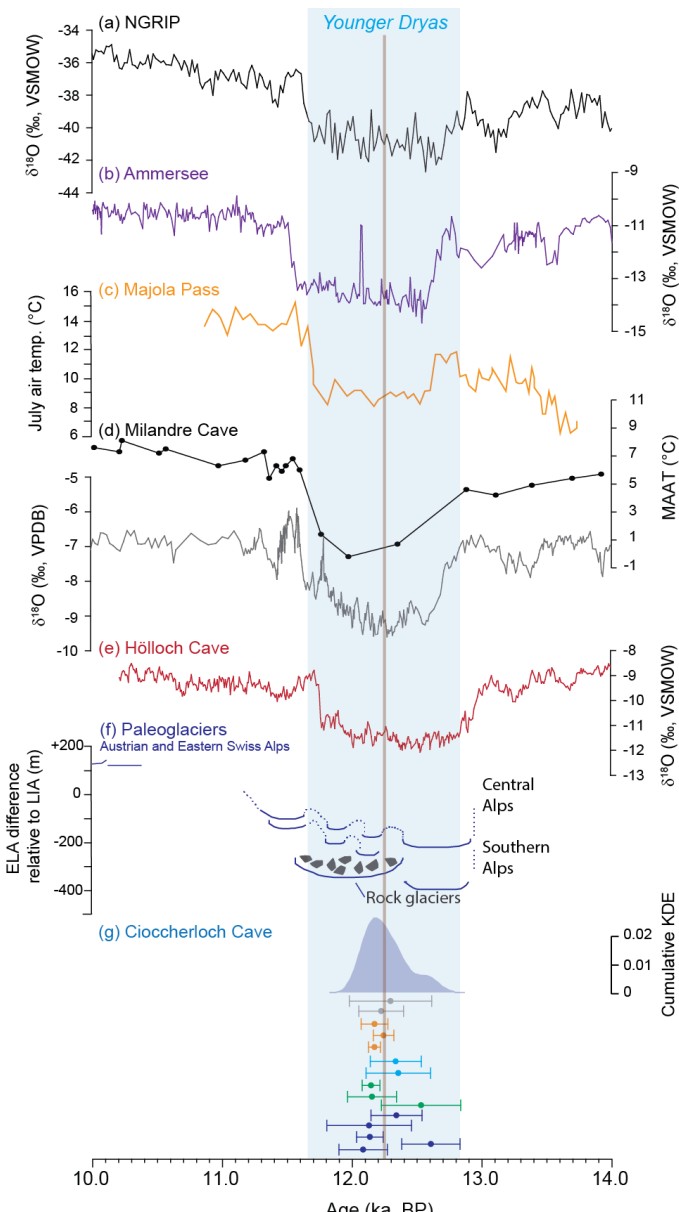

**Figure 6: CCC ages from Cioccherloch (g) compared to YD proxy records in Europe and Greenland plotted on their published chronology. (a) NGRIP δ¹⁸O data (Rasmussen et al., 2014), (b) lacustrine benthic ostracod δ¹⁸O data from Ammersee (von Grafenstein et al., 1999). YD paleotemperature reconstructions are shown by (c) chironomid-inferred July air temperatures from the Majola Pass (Ilyashuk et al., 2009) and by (d) MAAT inferred from fluid inclusion stable isotope data of stalagmites and speleothem δ¹⁸O data from Milandre Cave (Affolter et al., 2019). CCC ages are compared to speleothem δ¹⁸O data from Hölloch Cave and (f) extent of Central and Southern Alpine paleoglaciers (Heiri et al., 2014b). Calculated Kernel density funcion of all ²³⁰Th ages is shown by the blue shaded area (g). Dark blue and green refer to ²³⁰Th ages of CCCs from heap A and B, respectively (f). Data from heaps C, D and E are shown in light blue, orange, and grey, respectively (g). The brown vertical line mark the mid-YD transition recorded at Meerfelder Maar (Lane et al., 2013).**

## Tables

**Table 1: Input parameters for 1d heat conduction models simulating climate during the Allerød interstadial (1), the early YD (2a-2e) and the late YD at the study site (3a-3c). Snow ΔT ascribes the attenuation of the winter cold by the snowpack, whereas the resultant annual air temperature used as a boundary condition for the thermal model is described as mean the annual effective temperature (MAET). Modern day values are shown for comparison.**

| | Modern | Allerød | Early YD | | | | | | Late YD | | |
|---|---|---|---|---|---|---|---|---|---|---|---|
| | | Scenario 1 | Scenario 2a | Scenario 2b | Scenario 2c | Scenario 2d | Scenario 2e | Scenario 3a | Scenario 3b | Scenario 3c |
| T July [°C] | 11 | 9 | 7 | 8 | 8 | 8 | 8 | 8 | 8 | 8 |
| T January [°C] | -6 | -8 | -20 | -13 | -13 | -12 | -12 | -12 | -11 | -11 |
| MAAT [°C] | 2.5 | 0.5 | -6.5 | -2.5 | -2.5 | -2.0 | -2.0 | -2.0 | -1.5 | -1.5 |
| Snow ΔT [°C] | | - | - | - | 5 | - | 4.7 | 2.0 | - | 2.5 |
| MAET [°] | | - | - | - | -1.3 | - | -0.9 | -1.5 | - | -0.8 |
| Initial thermal conditions | | - | output of scenario 1 | output of scenario 1 | output of scenario 1 | output of scenario1 | output of scenario 1 | output of scenario 2e | output of scenario 2c | output of scenario 2c |

**Table 2: $^{230}$Th dating results of cryogenic calcite samples (FOS12) and a stalagmite (Cioc1) from Cioccherloch. Letters in the sample names of CCC (i.e. A, B, C, D, E) indicate the heap where the sample was collected, as shown in Fig. 2. Four samples were excluded from the discussion (italics) because of their excessive $^{232}$Th values. $\delta^{234}U = ([^{234}U/^{238}U]_{activity} - 1) \times 1000$. $\delta^{234}U_{initial}$ was calculated based on $^{230}$Th age (t). i.e. $\delta^{234}U_{initial} = \delta^{234}U_{measured} \times e^{\lambda 234 \times t}$. Ages are reported as BP. i.e. before the year 1950 AD. The error is 2 sigma.**

| Sample | $^{238}$U (ppb) | $^{232}$Th (ppt) | $^{230}$Th/$^{232}$Th (atomic x10$^{-6}$) | $\delta^{234}$U (measured) | $^{230}$Th/$^{238}$U (activity) | $^{230}$Th age (ka) (uncorr.) | $\delta^{234}$U$_{initial}$ (corr.) | 230Th age (ka) (corr.) |
|---|---|---|---|---|---|---|---|---|
| FOS12-A10 | 1883 ± 2 | 18024 ± 361 | 208 ± 4 | 121.9 ± 1.3 | 0.1208 ± 0.0006 | 12.40 ± 0.07 | 126 ± 1 | 12.08 ± 0.19 |
| *FOS12-A12* | *1139 ± 2* | *53468 ± 1074* | *46 ± 1* | *124.8 ± 2.2* | *0.1303 ± 0.0007* | *13.40 ± 0.08* | *129 ± 2* | *12.11 ± 0.86* |
| *FOS12-A3a* | *1863 ± 3* | *41219 ± 828* | *93 ± 2* | *125.0 ± 2.1* | *0.1252 ± 0.0006* | *12.83 ± 0.07* | *129 ± 2* | *12.20 ± 0.41* |
| FOS12-A3b | 1810 ± 3 | 20310 ± 408 | 185 ± 4 | 123.0 ± 2.1 | 0.1261 ± 0.0008 | 12.96 ± 0.09 | 128 ± 2 | 12.60 ± 0.22 |
| FOS12-A1 | 1480 ± 9 | 934 ± 21 | 3113 ± 70 | 121.4 ± 2.9 | 0.1191 ± 0.0009 | 12.22 ± 0.10 | 126 ± 3 | 12.13 ± 0.10 |
| FOS12-A2 | 1893 ± 19 | 29808 ± 666 | 128 ± 3 | 120.5 ± 4.4 | 0.1225 ± 0.0014 | 12.60 ± 0.16 | 125 ± 5 | 12.13 ± 0.33 |
| FOS12-A13 | 1466 ± 2 | 15563 ±312 | 192 ± 4 | 122.6 ± 1.5 | 0.1235 ± 0.0003 | 12.68 ± 0.03 | 127 ± 2 | 12.34 ± 0.20 |
| FOS12-B4 | 1492 ± 3 | 24543 ± 493 | 127 ± 3 | 124.0 ± 2.1 | 0.1268 ± 0.0004 | 13.02 ± 0.05 | 128 ± 2 | 12.53 ± 0.31 |
| *FOS12-B5* | *1790 ± 4* | *77634 ± 1564* | *53 ± 1* | *121.8 ±2.5* | *0.1382 ± 0.0006* | *14.30 ± 0.08* | *126 ± 3* | *13.11 ± 0.80* |
| FOS12-B6 | 1978 ± 2 | 6288 ± 126 | 622 ±13 | 122.4 ±1.4 | 0.1199 ± 0.0003 | 12.30 ± 0.04 | 127 ± 1 | 12.15 ± 0.07 |
| FOS12-B10 | 2095 ± 4 | 19878 ± 339 | 211 ± 4 | 121.1 ± 2.1 | 0.1213 ± 0.0006 | 12.47 ± 0.07 | 125 ± 2 | 12.15 ± 0.19 |
| FOS12-C | 1979 ± 4 | 25984 ± 522 | 156 ± 3 | 123.1 ± 2.0 | 0.1243 ± 0.0006 | 12.76 ± 0.06 | 127 ± 2 | 12.35 ± 0.25 |
| FOS12-C2 | 2050 ± 1 | 21760 ± 436 | 192 ±4 | 122.9 ± 1.5 | 0.1235 ± 0.0003 | 12.68 ± 0.03 | 127 ± 2 | 12.34 ± 0.20 |
| FOS12-D | 1782 ± 1 | 6916 ± 139 | 514 ± 10 | 122.8 ± 1.3 | 0.1210 ± 0.0003 | 12.41 ± 0.04 | 127 ± 1 | 12.24 ± 0.08 |
| *FOS12-D3* | *1458 ± 3* | *47447 ± 953* | *61 ± 1* | *121.9 ± 2.0* | *0.1212 ± 0.0005* | *12.44 ± 0.06* | *126 ± 2* | *11.53 ± 0.60* |
| FOS12-D9 | 2189 ± 2 | 3975 ± 80 | 1087 ± 22 | 121.4 ±1.4 | 0.1197 ± 0.0003 | 12.29 ± 0.03 | 126 ± 1 | 12.17 ± 0.05 |
| FOS12-D10 | 1760 ± 2 | 8134 ± 160 | 430 ± 9 | 123.4 ± 1.6 | 0.1206 ± 0.0005 | 12.36 ± 0.06 | 128 ± 2 | 12.17 ± 0.10 |
| FOS12-E | 826 ± 1 | 6841 ± 137 | 242 ± 5 | 121.5 ± 1.3 | 0.1217 ± 0.0007 | 12.50 ± 0.08 | 126 ± 1 | 12.22 ± 0.17 |
| FOS12-E2 | 2026 ± 3 | 34463 ± 693 | 121 ± 2 | 121.2 ± 1.9 | 0.1244 ± 0.0005 | 12.80 ± 0.06 | 126 ± 2 | 12.29 ± 0.32 |
| Cioc1-1 | 47 ± 0.1 | 396 ± 8 | 34 ± 5 | 177.4 ± 3.3 | 0.0171 ± 0.0024 | 1.59 ± 0.23 | 178 ± 3 | 1.32 ± 0.27 |
| Cioc1-2 | 47 ± 0.1 | 293 ± 6 | 175 ± 5 | 200.1 ± 2.8 | 0.0654 ± 0.0012 | 6.10 ± 0.11 | 203 ± 3 | 5.88 ± 0.15 |
| Cioc1-3 | 46 ± 0.1 | 80 ± 2 | 1570 ± 36 | 273.1 ± 2.3 | 0.1653 ± 0.0014 | 15.09 ± 0.14 | 285 ± 2 | 14.98 ± 0.14 |