# Peer review of "Cryogenic cave carbonates in the Dolomites (Northern Italy): insights into Younger Dryas cooling and seasonal precipitation"

_Climate of the Past, 2020_

## Referee Comment (RC1) · Anonymous Referee #1 · 26 Sep 2020

This is an interesting manuscript investigating the internal climatic structure of YD, which in my opinion warrants publication, but I do think further reflection is probably in order. Since much of the paper's discussions/conclusions are drawn from the results of the thermal modeling experiments, it would be helpful if the method is presented in more details (maybe some references would also be appropriate). This could be done by adding additional text in the manuscript or supplemental material section.

The attempt to characterize the early YD climate, basically using only two (at most three) dated CCC may be a little too far reaching, especially given their associated errors. I think authors need to present more convincing evidence for their early YD

discussion since three of the scenarios (2b to 2d) are somehow marginally supported by only two CCC samples. This situation is in great contrast with the late YD, for which ~7 dated samples exist. Better explaining why and how scenarios 2b to 2d are really relevant to the discussion would be helpful.

Authors are using various input parameters for their thermal modeling and end up presenting MAAT, $\Delta$MAAT, MAET, $\Delta$T, snow $\Delta$T, etc., point at which tracking all these values in sections 4.4.1 to 4.4.3 is rather difficult and easy to mix up digits. Furthermore, they don't always match with what is reported in Table 2 or figures caption (e.g., scenario 2d and 2e is said to be forced with a MAAT of -1.5°C in the caption of figure 3 (line 534), but in Table 2 it appears to be -2°C; line 535 reads "... $\Delta$T of 4.5°C" for scenario 2e, but in Table 2 the value is 4.7°C. The presentation of data in these sections needs to ve revised and made clearer. One way would be to add all values use in Table 2 so that is easier to track them. On the same vein, I see authors derived and reported in Table 2 the mean annual effective temperature, but nowhere in text these values are discussed.

I found Figure 3 to be rather difficult to understand. Some minor improvements, such as placing "early YD" in the right side of the plot and making the blue dashed line more visible, would certainly improve it. However, as expressed above, it is unclear to me which of the CCC really characterize scenarios 2b, c, and d, as I see only two ages with 400 to 600 yrs error that could be assigned to early YD.

I also have many small comments and suggestions that I think would improve the language and clarity of the manuscript.

I suggest using throughout the text, figures and captions capital letter C in Cave when is a proper name (e.g., Hölloch Cave, Milandre Cave, etc.).

Line 60: what do you mean by "certain proxy properties"?

Line 67: Authors could probably make use of the recently published study of Cheng et

al. 2020 in PNAS

Line 77: CCC are in fact speleothems not cave sediments, thus, I urge authors to consider them as such.

Line 84: maybe "CCC form in caves with perennial ice. . ." will be more clearer to readers than "CCC form within perennial cave ice . . ."

Line 105: Methods - is there any other way of presenting the information in this chapter without breaking it so heavily and have only 2-3 lines for various sub-chapters?

Line 129: Thermal modeling - additional information and references are needed in order to better understand the method (e.g., what might be the effect of taking 0.5 or 1 for dT/dz? What are the uncertainties of the results associated with this model?)

Line 131: a reference to whoever generated the heat flow model would be appropriate

Line 135 Equation 1 - if authors consider dT/dz = 0 then Q is 0 regardless of thermal conductivity, right? Do I miss something on how this equation really helps?

Line 136: thermal diffusivity (how fast heat diffuses through a material) is not the same with thermal conductivity (ability of a system to transport heat energy). Authors define thermal conductivity as "c" in Equation 1, but then set the thermal diffusivity of limestone to 1.2x1010^-6. What value was actually used for Eq. 1, which once again, if dT/dz is assumed 0, Q would be 0.

Line 150 - 153: For consistency, use XX‰ or XX ‰ but not both ways.

Lines 180-223: I feel that the presentation of these scenarios could be clearer if all values are included in Table 2 or those already in this table are presented in text as well. Right now, I see, for some scenarios, different values in text, table, and/or caption of figures 3 and 4.

Lines 274-275: what do you mean by "Scenario 2e including provides . . ."

Line 301: it was not immediately clear to this reviewer how the value of 5.7°C was derived. Please add text to clarify.

Lines 346-347: my suggestion for rewording this part of the sentence: "... CCC in the Dolomites, which in contrast to many studies from Central European caves, formed not during ..."

Line 358: add "for" at "advocates for a mild..."

Line 542 - Figure 4 - winter snow cover ($\Delta T = 2$°C) is mentioned in this caption, but it is not in Table 2 for scenario 3a.

─────────────────────────

---

## Referee Comment (RC2) · Anonymous Referee #2 · 25 Oct 2020

Koltai et al. use a combination of cave air temperature modeling and U/Th dating of cryogenic cave carbonates (CCC) to discuss climate variability during the YD in the SE Alps (Dolomites). The topic is becoming a hotly debated one, as more and more studies suggest that climate during the YD was spatially, temporally and seasonally different throughout Europe and the author's paper comes to add more information to this debate. While bot the title and discussion are tantalizing, I fond that the authors overstretch themselves in analyzing to many climate variables over a long period of time based on a limited data set (several U/Th dates) buttressed by modeling. Several points should be made clear before publication of the paper. I detail these in a few general and technical comments below.

[Figure]

Several papers discussed climatic inferences based on the U/Th age of CCCs and a wide range growth periods have been found leading to several possible climatic conditions leading to the precipitation of CCC (Zak et al., 2004, 2009, 2012, Luetscher et al., 2013, Spoetl and Cheng, 2014 – quite a few of these are missing from the cited literature section...). These authors have found CCS growing during warm and cold, dry and wet periods during MIS6, MIS4, MIS3, MIS 2, mid-to-late Holocene (Roman and Medieval Warm Periods). From these studies, it occurs that a wide range of external climatic conditions are possibly favourable for the formation of CCCs in caves and it is the peculiarities of cave climate that are in the end responsible for this. Consequently, I find the climatic inferences made in this paper somehow only poorly supported by the data but strongly relying on the thermal modeling. While the data are what they are, the modeling methods and results should be explained in more detail and the various assumptions (e.g., lack or presence of permafrost, assumed temperatures, buffering effect of snow cover etc) in choosing input data and favouring one model over the other better explained. Further, the authors could summarize the climatic conditions during the YD in a simplified figure, emphasizing the seasonally distinct climatic conditions and the two-part YD climate and than add their data in support of the inferred climatic conditions. The concluding figure 6 does not clearly supports the authors' claims.

Specific comments:

28 – GS1 starts at 12.9 ka, not 12.8 ka (Rasmussen et al., 2014)

35 – catastrophic is rather human-centered

37 – perhaps "cold" is enough, Siberian-like is quite subjective (and given that this is a paleoclimate paper, Siberian climate varied widely in the past)

41 – relative to...?

48-52 – to which season do these reconstructions refer?

69-70 – not clear how "enhanced precipitation differences between the northern, central and southern part of the Alps" would result in a YD maximum. Also, check the 13.5 ka age, it is well before the onset of the YD

73 – what do you mean by "double response"? Two periods of glacier advance? Please clarify

86-88 – I particularly enjoy this statement, but please clarify 1) what do you mean by "strong winter", 2) what season the "1-2 C warming" refers to and 3) the reference for the 'drier' comparative (e.g., "drier" compared to early YD?)

91 – This sentence is a odds with the cave's description here

103-104 – over what period were these snow depth values measured? Snowfall heights do not record the amount if snowfall accurately, please provide the total amount of winter and early winter (September-December) precipitation.

114 – What was exactly sampled for stale isotope analyses? Entire CCC? Outer/inner part f it? Please detail.

120 – I understand that these CCCs grow over prolonged periods of time. What part of the individual CCCs was sampled for dating? Or was it whole sample?

200 – why was a 5 °C temperature chosen for the buffering effect of snow cover?

230 – how long does it take for these CCCs to form? Several years is not that much in terms of YD climate variability, so with only 3 ages for the early YD, the inferences made in this article might be slightly far-fetched

235-236 – here is a bit of a jump in logic, as a few lines above (230) a few years are required for CCC to form and now the suggestion is that cave climate was stable for centuries

245 – do you have any indication on when ruble closed the connecting gallery? It could have been open during the YD and hence the cave would have been out of thermal equilibrium with the outside, as discussed above

246-248 – I would argue that CCC record changes in the thermal state of the cave, that could be or not in equilibrium with external conditions. The assumption is that the ruble blocking the cave was there throughout and since the YD

250-251 – Liquid water reaching the cave would have a dramatic impact on temperature, given the extremely high specific heat capacity of water. I think this is to easily dismissed. And if I understood right, liquid water was required to form CCC (line 259 in the text)

264 – U/Th ages show that CCC formed well (several hundreds of years) into the YD, not at the transition.

274 – how likely is that the cave was perennially frozen throughout both winters and summers for several centuries? If not, than the proposed shielding by snow is not required to induce changes in cave air temperature around 0 and thus facilitate the precipitation of CCC

277 and subsequent discussion – winter or summer very cold YD? The distinction was heavily promoted in the introduction, it should be made here, as well.

289 – see my comment on the ruble blocking the cave and its effect on cave climate. How was the 3 C temperature obtained?

292 and subsequent –warm YD summers indicate that the cave would have been warm enough to lead to ice melting and prevent the all-over freezing of cave. Consequently, CCC could have precipitated by the freezing of water formed during warm summers on the surface of (cold winter forming) cave ice. Is this a likely scenario?

303-305 – this shielding would not be required if the scenarios presented above could be happening.

320 – U/Th ages indicate that CCC formed for several centuries, so why the emphasize on this correlation with the mid-YD transition? The age errors and the widespread ages of all CCCs are to large compared to the narrow age of the transition to support the

subsequent discussion.

---

## Short Comment (SC1) · 25 Oct 2020

From carefully reading this very interesting study, it appears that the results provide important insights on the temporal changes in the amount and timing of snow fall at the study site during autumn and early winter during the YD. Koltai et al. estimate rather warm January temperatures of around -13.7°C for the YD and only a quite moderate change in seasonality of up to 5.7K relative to the AL. Based on these results, the authors "challenge the commonly held view of extreme YD seasonality" (e.g. line 15).

While the inference of changes in snow cover over time are important by itself, I do not agree that the results can challenge seasonality changes from other studies: The

Interactive
comment

inference of snow-rich conditions and that such a snow cover insulates the cave cannot be used with much confidence to estimate the full severity of winter temperatures and hence can neither reconstruct nor challenge seasonality changes in a general way.

In the best case, the results may be valid for the local cave or regional climate setting. However, the authors do not provide evidence for why the results from a cave record at a high elevation from the Southern Alps can generally challenge commonly held views on extreme YD seasonality in other regions and, i.e., not north of the Alps across the Euro-Atlantic region.

As discussed by the authors, the timing and amount of snow cover over the cave has a major control on subsurface temperature changes. An early and/or thick snow cover will protect the cave from the most severe winter cooling like in January. It is therefore quite likely that the cave record fails to estimate the full winter cooling which would define the amplitude of seasonality change (this might also apply to other studies affected by ice or snow cover insulation). This would imply that the authors do not reconstruct the full seasonality. Hence, they cannot challenge seasonality results from other regions. It is unclear to me how assumptions in the thermal modelling can account for the combination of two unknowns: an unknown winter severity in the YD together with an unknown snow thickness. I think this should be clarified in the text.

Line 37: "Siberian-like" would imply extreme seasonality which is typical for continental climates. Such a climate is reasonable for the YD north of the Alps as the major heat source is shut off with an ice-covered North Atlantic Ocean. This does not need to apply to the Southern Alps, though.

Line 39: Remove "however" as the sentence before is about winter and this sentence is about summer.

Line 39-40: Replace "shutdown" with slowdown - the study assumes an AMOC slowdown of around 36% relative to the Alleröd. The 4.3 to 0.3 summer cooling refers to chironomid-based estimates and hence a notable summer cooling – the mild YD summers are based on plant indicator species and the climate model which suggest no overall summer cooling.

Line 49: The -10 K change in the northern Alps would be an indication that your record from the Southern Alps does not reflect the same changes – hence you cannot challenge large seasonality changes in general. If anything, your study reflects local or even only cave ambient temperatures/snow cover changes and additional information is required to claim these muted changes would generally apply to south of the Alps. Spagnolo & Ribolini (2019, see section 4.3) estimate a seasonality of 21 degrees for the maritime Alps at the ELA during the YD compared to 14.2 degrees today.

Lines 62-63: While it is true that Hepp et al. 2019 claim the opposite, several comments by E. Schefuss, B. Zollitschka as well as D. Sachse and myself (see the online discussion: https://cp.copernicus.org/articles/15/713/2019/cp-15-713-2019-discussion.html) raise serious questions regarding the validity of that study. The extensively studied Meerfelder Maar nearby does not agree at all with the interpretation using a much more reliable chronology. I think this disagreement needs to be at least mentioned here (e.g. relative to Brauer et al., 1999; Bakke et al., 2009).

Line 87: Again, there is no reason why the local cave record should "argue against strong winter cooling during the early YD" at other places and perhaps even not at the site itself.

345-350: This might be true for the local setting in the cave insulated by a snow cover but might fail to reconstruct the full MAAT cooling which happened in air temperatures above the snow cover.

Bakke J., Lie Ø., Heegaard E., Dokken T., Haug G. H., Birks H. H., Dulski P. and Nilsen T. (2009): Rapid oceanic and atmospheric changes during the Younger Dryas cold period. Nat Geosci 2, 202–205.

Brauer A., Endres C., Gunter C., Litt T., Stebich M. and Negendank J. (1999): High

resolution sediment and vegetation responses to Younger Dryas climate change in varved lake sediments from Meerfelder Maar, Germany. QSR 18, 321–329.

Spagnolo, M and A. Ribolini (2019): Glacier extent and climate in the Maritime Alps during the Younger Dryas. Palaeogeography, Palaeoclimatology, Palaeoecology, 536: 109400, https://doi.org/10.1016/j.palaeo.2019.109400

---

## Author Comment (AC2) · 11 Nov 2020

Submission of reply to the comments made by Reviewer #2
Ms. Ref. No.: CP-2020-107
Title: Cryogenic cave carbonates in the Dolomites (Northern Italy): insights into Younger Dryas cooling and seasonal precipitation

**Reviewer #2**

We thank the reviewer for his/her critical comments. We address below the points raised by this referee (in italics) and try to clarify what we are doing in response to these comments (blue).

Sincerely,

On behalf of all the co-authors,
Gabriella Koltai

_Main comments_

_Koltai et al. use a combination of cave air temperature modeling and U/Th dating of cryogenic cave carbonates (CCC) to discuss climate variability during the YD in the SE Alps (Dolomites). The topic is becoming a hotly debated one, as more and more studies suggest that climate during the YD was spatially, temporally and seasonally different throughout Europe and the author's paper comes to add more information to this debate. While bot the title and discussion are tantalizing, I fond that the authors overstretch themselves in analyzing to many climate variables over a long period of time based on a limited data set (several U/Th dates) buttressed by modeling. Several points should be made clear before publication of the paper. I detail these in a few general and technical comments below._

The description of our approach may have been unclear and we are thankful for pointing this out. This paper utilizes a novel paleoclimate archive (cryogenic cave carbonates, or CCCs for short) that allows to precisely constrain permafrost thawing events in the past, when the cave air temperature was very close to the freezing point (e.g. Zak et al., 2018 and references therein, for further discussion on CCC formation see lines 76-85 and 225-266).

In our study we use a 1-d heat conduction model developed by one of the co-authors to investigate how the atmospheric climate signal is transferred into the subsurface. The different scenarios are based on regional proxy data reconstructions for the Allerød (Ilyashuk et al., 2009a) and the YD (e.g. Affolter et al., 2019; Frauenfelder et al., 2001; Ghadiri et al., 2018; Luetscher et al., 2015). These studies suggest an approximately 3 to 10°C decrease in mean annual air temperature (MAAT) during the YD compared to modern day. Our experiments take advantage of these studies and investigate under which climate conditions CCCs could have formed at our study site.
The heat-flow model simulates the penetration of the ambient seasonal temperature signal to 50 m depth. We use local meteorological data to characterize modern day conditions (see Table 2) and palaeotemperature estimates from the Majola Pass (Ilyashuk et al., 2009b) to define the input parameters for scenario 1 (Allerød interstadial climate). As a recent study by Schenk et al. (2018) suggested that YD summers remained relatively warm with temperature decreases of 4.3°C in NW Europe and 0.3°C in E Europe relative to the preceding Bølling interstadial, we kept the July temperatures 3-4°C lower modern values (Table 2) and attributed most of the MAAT change to

winter cooling. We use the output of this simulation as the starting condition for all early YD experiments. As a second step, we model the penetration of the seasonal signal without the presence of winter snow to provide an endmember for the YD cooling (scenarios 2a, 2b, 2d). The results show that the subsurface would be overcooled and prevent CCC formation latest 100 years after the start of the cooling. Then as a next step we include the buffering effect of a winter snowpack insulating the ground from the winter chill. This buffering effect (snow $\Delta T$) is set to its maximum in scenarios 2c and 2e to test if a similar amplitude of cooling investigated in scenarios 2b and 2d (Table 2) would allow CCC formation given the presence of a winter snow cover. As discussed in the manuscript (lines 304-306) studies in modern permafrost areas suggest that even a 35 cm thick stable winter snow cover may result in a 5.5°C increase in mean ground surface temperatures (Zhang, 2005 and references therein). Therefore, our snow $\Delta T$ values of 5°C and 4.7°C are considered to be realistic for the YD at this alpine setting. With these two input parameters we characterize the maximum possible amplitude of winter cooling in the absence or presence of a winter snow cover.

*Several papers discussed climatic inferences based on the U/Th age of CCCs and a wide range growth periods have been found leading to several possible climatic conditions leading to the precipitation of CCC (Zak et al., 2004, 2009, 2012, Luetscher et al., 2013, Spoetl and Cheng, 2014 – quite a few of these are missing from the cited literature section. . .). These authors have found CCS growing during warm and cold, dry and wet periods during MIS6, MIS4, MIS3, MIS 2, mid-to-late Holocene (Roman and Medieval Warm Periods). From these studies, it occurs that a wide range of external climatic conditions are possibly favourable for the formation of CCCs in caves and it is the peculiarities of cave climate that are in the end responsible for this. Consequently, I find the climatic inferences made in this paper somehow only poorly supported by the data but strongly relying on the thermal modeling. While the data are what they are, the modeling methods and results should be explained in more detail and the various assumptions (e.g., lack or presence of permafrost, assumed temperatures, buffering effect of snow cover etc) in choosing input data and favouring one model over the other better explained.*

We agree with the reviewer that the local cave microclimate may influence CCC formation to a considerable extent in complex cave systems (cf. also Koltai et al., 2020), however we are confident that this was not the case during the YD in Cioccheroch Cave. In the manuscript (lines 96-101) we present the temperature data of a 1-year monitoring of the CCC-bearing cave chamber and provide a discussion the microclimate of the site (lines 238-253).

Regarding the modeling methods, we refer to our previous comment from Reviewer#2 (. We would like to emphasize that all previous studies interpreted CCCs as proxies for paleo-permafrost thawing. In our paper we go a step further and apply a 1-d heat flow model to characterize under which climate conditions CCCs could have formed in the cave. We emphasize in the manuscript that CCCs form under a stable cave microclimate when the cave air temperature is negative and very close to the 0°C isotherm. If the cave would have been strongly ventilated and heat advection played an important role, the fine crystalline variety of CCCs (essentially crystal powders) would have formed instead of the coarse crystalline one. The innovative aspect of our study is the quantitative link between the climate signal recorded by the CCCs to the surface environment.

The references will be added to the manuscript.

*Further, the authors could summarize the climatic conditions during the YD in a simplified figure, emphasizing the seasonally distinct climatic conditions and the two-part YD climate and than add their data in support of the inferred climatic conditions. The concluding figure 6 does not clearly supports the authors' claims.*

We are not sure why reviewer feels that Fig. 6 is not clearly supporting our claims. We believe that Fig. 5 summarizes these climatic conditions obtained by the heat-flow modeling experiments. We nevertheless appreciate the reviewer´s suggestion and will try to improve this figure.

*Specific comments*

*28 – GS1 starts at 12.9 ka, not 12.8 ka (Rasmussen et al., 2014)*

The start of GS1 was at 12.896 yr b2k (Rasmussen et al., 2014), which is 12.846 yr BP.

*35 – catastrophic is rather human-centered*

We will change this.

*37 – perhaps "cold" is enough, Siberian-like is quite subjective (and given that this is a paleoclimate paper, Siberian climate varied widely in the past)*

We will rephrase this.

*41 – relative to. . .?*

Relative to the Bølling-Allerød interstadial (see line 41)

*48-52 – to which season do these reconstructions refer?*

These reconstructions refer to annual air temperature as stated in line 50.

*69-70 – not clear how "enhanced precipitation differences between the northern, central and southern part of the Alps" would result in a YD maximum. Also, check the 13.5 ka age, it is well before the onset of the YD*

We will rephrase this sentence for clarification. A new reference will be added for the 13.5 ka BP age (Ivy-Ochs, 2015).

*73 – what do you mean by "double response"? Two periods of glacier advance? Please clarify*

Yes, we mean two glacier advances as reported by Baroni et al. (2017).

*86-88 – I particularly enjoy this statement, but please clarify 1) what do you mean by "strong winter", 2) what season the "1-2 C warming" refers to and 3) the reference for the 'drier' comparative (e.g., "drier" compared to early YD?)*

This will be rephrased in the revised manuscript.
For clarification
    (1) by strong winter cooling we refer to the 10°C decrease in MAAT compared to present day, as reported from the Jura Mountains (Affolter et al., 2019; Ghadiri et al., 2018), which requires a disproportionally large winter cooling considering that the summer cooling for this part of Europe was rather small as shown by pollen and chironomid data (as detailed in the ms.).
    (2) the 1-2°C warming refers to the MAAT
    (3) and drier is meant relative to the early YD (as stated in line 89)

*91 – This sentence is a odds with the cave's description here*

We will remove this sentence.

*103-104 – over what period were these snow depth values measured? Snowfall heights do not record the amount if snowfall accurately, please provide the total amount of winter and early winter (September-December) precipitation.*

Snow height is monitored in the Dolomites at Rossalm and Piz la Ila stations. Unfortunately, the total amount of precipitation is not measured. We used the monthly data for the last seven years (2012-2019) from Rossalm and for a fifteen-year-observation-period (1999-2014) at Piz la Ila to calculate average snowfall amounts for the autumn (September to December).

*114 – What was exactly sampled for stale isotope analyses? Entire CCC? Outer/inner part f it? Please detail.*

Most of the CCCs were too tiny to be cut therefore a handheld drill was used to take small aliquots of carbonate powder for stable isotope analyses. Usually, the outer layer was drilled off and discarded and then the carbonate powder was drilled 1-2 mm below the surface. In case of the dated CCC samples, a small aliquot of the drilled carbonate powder was used for stable isotope analyses.

*120 – I understand that these CCCs grow over prolonged periods of time. What part of the individual CCCs was sampled for dating? Or was it whole sample?*

This is partially explained in the manuscript (line 120-122), and more details will be provided in the revised manuscript. If the single crystals were large enough, the carbonate powder was drilled from the center of the CCC to define the start of CCC formation (15 samples). In case of two skeletal crystals, the entire crystal was used for $^{230}$Th dating.

*200 – why was a 5 _C temperature chosen for the buffering effect of snow cover?*

Please see our response above.

*230 – how long does it take for these CCCs to form? Several years is not that much in terms of YD climate variability, so with only 3 ages for the early YD, the inferences made in this article might be slightly far-fetched*

The reviewer asks a long-standing question in the community working on CCCs. The short answer is that we only know that the fine crystalline variety of CCCs (essentially crystal powder) forms within a matter of hours to days by comparably rapid freezing of a water film. The coarse crystalline CCC variety – the one we talk about in this study – has never been observed in statu nascendi, and nobody has made experiments growing them under controlled conditions. Still, there is unanimous consensus among colleagues working on coarse crystalline CCC that these carbonates form (a) not within a water film but in freezing pools in the ice, and (b) require much longer to form than their fine crystalline counterparts. This is pretty obvious given the well-developed macroscopic crystals and the fact that in other caves CCC can reach several cm in diameter. There is no published information how much time is involved in the formation of individual coarse crystalline CCC aggregates. In our group we also study Pleistocene CCCs in Siberian caves and they can be up to a few cm in diameter. We dated core and rim of these large CCCs. Even the most precise $^{230}$Th ages cannot resolve an age difference within individual CCCs. Considering the age uncertainties up to several hundred years could be involved in the growth history of some of these aggregates. Although the Cioccherloch samples are significantly smaller than their counterparts from Siberia we are convinced that months to years (and decades in the case of larger particles) are likely involved in their formation. These assumptions are also supported by the smooth isotope and trace element distribution patterns of these particles.

We disagree with the reviewer on the inferences being too far-fetched. During the review process we dated two more CCCs from heaps A and C. These analyses yielded $^{230}$Th ages of 12.34±0.2 ka and 12.33±0.2 ka BP, providing further support for CCC formation during the early YD. We strongly believe that neglecting the possibility of early YD CCC formation (as supported by the $^{230}$Th ages and their 2σ uncertainties) would be an oversimplification.

*235-236 – here is a bit of a jump in logic, as a few lines above (230) a few years are required for CCC to form and now the suggestion is that cave climate was stable for centuries.*

We do not see a problem here, and please see our comment above on CCC formation. The majority of $^{230}$Th ages overlap within their 2σ errors. Similar prolonged periods of CCC formation have been reported form other alpine caves (e.g. Luetscher et al., 2013; Spötl and Cheng, 2014, Spötl et al., in review)

*245 – do you have any indication on when ruble closed the connecting gallery? It could have been open during the YD and hence the cave would have been out of thermal equilibrium with the outside, as discussed above*

We have no information on when the rubble closed the narrow connection.

*246-248 – I would argue that CCC record changes in the thermal state of the cave, that could be or not in equilibrium with external conditions. The assumption is that the ruble blocking the cave was there throughout and since the YD*

We disagree. The presence of a snow cone slightly larger than today could have also sealed this connection. Also, the connection between the CCC-bearing chamber and the entrance shaft could not have been more open in the past than it is today. We provide a discussion on advective processes in lines 283-298.

*250-251 – Liquid water reaching the cave would have a dramatic impact on temperature, given the extremely high specific heat capacity of water. I think this is to easily dismissed. And if I understood right, liquid water was required to form CCC (line 259 in the text)*

We considered this process qualitatively and concluded that drip water obviously entered this cave chamber forming meltwater pools on the ice, eventually giving rise to CCC formation. This very slow flowing seepage water, however, is likely thermally equilibrated with the ca. 50 m-thick rock above the cave and given its very low discharge carries comparably little heat from the surface. In addition, the YD climate was likely drier than today (as discussed in the ms.) hence discharge was even lower.
We will expand this section to make this point clearer.

*264 – U/Th ages show that CCC formed well (several hundreds of years) into the YD, not at the transition.*

This sentence will be rewritten.

*274 – how likely is that the cave was perennially frozen throughout both winters and summers for several centuries? If not, than the proposed shielding by snow is not required to induce changes in cave air temperature around 0 and thus facilitate the precipitation of CCC*

CCC only form very close to 0°C and our data therefore rule out that this cave was well below 0°C during the YD.

*277 and subsequent discussion – winter or summer very cold YD? The distinction was heavily promoted in the introduction, it should be made here, as well.*

A separate subsection is devoted to the discussion of seasonality changes (5.3 Increased seasonality in the early YD, lines 289-313)

*289 – see my comment on the ruble blocking the cave and its effect on cave climate. How was the 3 C temperature obtained?*

This maximum amplitude cooling (≤3°C) at the Allerød-YD transition was derived from thermal modeling (scenarios 2c and 2e, please see lines 272-276).

*292 and subsequent –warm YD summers indicate that the cave would have been warm enough to lead to ice melting and prevent the all-over freezing of cave. Consequently, CCC could have precipitated by the freezing of water formed during warm summers on the surface of (cold winter forming) cave ice. Is this a likely scenario?*

That is an interesting suggestion. Warm summers are accounted for in the modeling experiments (Scenarios 2a-2e). However, we the seasonal temperature signal is cancelled out after the first 10 meters of rock (see the figures showing the heat-flow model results).

CCCs cannot form by repeated thaw-freeze cycles of cave ice as there would not be enough dissolved ions in the water to reach supersaturation via freezing and to precipitate cryogenic minerals. Drip water derive from the epikarst and the vadose zone is needed to deliver the solutes necessary for CCC formation. As slow freezing proceeds, the expulsion of these ions leads to supersaturation and consequently the precipitation of CCCs (e.g. Žák et al., 2012, 2008). This is corroborated by U concentrations much higher than those of warm-climate, non-cryogenic speleothems from the same cave (see Table 1 in the ms.).

*303-305 – this shielding would not be required if the scenarios presented above could be happening.*

Please see our comment above.

*320 – U/Th ages indicate that CCC formed for several centuries, so why the emphasize on this correlation with the mid-YD transition? The age errors and the widespread ages of all CCCs are to large compared to the narrow age of the transition to support the subsequent discussion.*

The majority of the [230]Th ages cluster at 12.2 ka BP. The weighted mean of all ages is 12.19±0.6 ka BP. As this coincides with the mid-YD transition (12.24 ±0.4 ka BP, see Lane et al., 2013) CCC formation in Cioccherloch Cave was likely connected to this climate event.

**References**

Affolter, S., Häuselmann, A., Fleitmann, D., Edwards, R.L., Cheng, H., Leuenberger, M., 2019. Central Europe temperature constrained by speleothem fluid inclusion water isotopes over the past 14,000 years. Sci. Adv. 5, eaav3809. https://doi.org/10.1126/sciadv.aav3809

Baroni, C., Casale, S., Carturan, L., Seppi, R., 2017. Double response of glaciers in the Upper Peio Valley ( Rhaetian Alps , Italy ) to the Younger Dryas climatic deterioration. https://doi.org/10.1111/bor.12284

Frauenfelder, R., Haeberli, W., Hoelzle, M., Maisch, M.A.X., 2001. Using relict rockglaciers in GIS-based modelling to reconstruct Younger Dryas permafrost distribution patterns in the Err-Julier area , Swiss Alps 55, 195–202. https://doi.org/10.1080/00291950152746522

Ghadiri, E., Vogel, N., Brennwald, M.S., Maden, C., Häuselmann, A.D., Fleitmann, D., Cheng, H., Kipfer, R., 2018. Noble gas based temperature reconstruction on a Swiss stalagmite from the last glacial–interglacial transition and its comparison with other climate records. Earth Planet. Sci. Lett. 495, 192–201. https://doi.org/10.1016/j.epsl.2018.05.019

Ilyashuk, B., Gobet, E., Heiri, O., Lotter, A.F., van Leeuwen, J.F.N., van der Knaap, W.O., Ilyashuk, E., Oberli, F., Ammann, B., 2009a. Lateglacial environmental and climatic changes at the Maloja Pass, Central Swiss Alps, as recorded by chironomids and pollen. Quat. Sci. Rev. 28, 1340–1353. https://doi.org/10.1016/j.quascirev.2009.01.007

Ilyashuk, B., Gobet, E., Heiri, O., Lotter, A.F., van Leeuwen, J.F.N., van der Knaap, W.O., Ilyashuk, E., Oberli, F., Ammann, B., 2009b. Lateglacial environmental and climatic changes at the Maloja Pass, Central Swiss Alps, as recorded by chironomids and pollen. Quat. Sci. Rev. 28, 1340–1353. https://doi.org/10.1016/j.quascirev.2009.01.007

Ivy-Ochs, S., 2015. Glacier variations in the European Alps at the end of the last glaciation 41, 295–315. https://doi.org/10.18172/cig.2750

Lane, C.S., Brauer, A., Blockley, S.P.E., Dulski, P., 2013. Volcanic ash reveals time-transgressive abrupt climate change during the Younger Dryas. Geology 41, 1251–1254.

https://doi.org/10.1130/G34867.1

Luetscher, A.M., Hellstrom, J., Müller, W., Barrett, S., 2015. Title : A strong seasonality shift during the Younger Dryas cold spell in the European Alps 1–27.

Schenk, F., Väliranta, M., Muschitiello, F., Tarasov, L., Heikkilä, M., Björck, S., Brandefelt, J., Johansson, A. V., Näslund, J.-O., Wohlfarth, B., 2018. Warm summers during the Younger Dryas cold reversal. Nat. Commun. 9. https://doi.org/10.1038/s41467-018-04071-5

Žák, K., Onac, B.P., Perşoiu, A., 2008. Cryogenic carbonates in cave environments: A review. Quat. Int. 187, 84–96. https://doi.org/10.1016/j.quaint.2007.02.022

Žák, K., Richter, D. K.Filippi, M., Živor, R., Deininger, M., Mangini, A., Scholz, D., 2012. Coarsely crystalline cryogenic cave carbonate – a new archive to estimate the Last Glacial minimum permafrost depth in Central Europe. Clim. Past 8, 1821–1837. https://doi.org/10.5194/cp-8-1821-2012

Zhang, T., 2005. Influence of the seasonal snow cover on the ground thermal regime: an overview. Rev. Geophys. 43, RG4002. https://doi.org/10.1029/2004RG000157

---

## Author Comment (AC3) · 11 Nov 2020

Submission of reply to the comments made by Frederic Schenk
Ms. Ref. No.: CP-2020-107
Title: Cryogenic cave carbonates in the Dolomites (Northern Italy): insights into Younger Dryas cooling and seasonal precipitation

**Dear Frederic Schenk,**

We appreciate your critical feedback on our manuscript and address below the points raised by you (in italics).

Sincerely,
On behalf of the co-authors,
Gabriella Koltai

*From carefully reading this very interesting study, it appears that the results provide important insights on the temporal changes in the amount and timing of snow fall at the study site during autumn and early winter during the YD. Koltai et al. estimate rather warm January temperatures of around -13.7_C for the YD and only a quite moderate change in seasonality of up to 5.7K relative to the AL. Based on these results, the authors "challenge the commonly held view of extreme YD seasonality" (e.g. line 15). While the inference of changes in snow cover over time are important by itself, I do not agree that the results can challenge seasonality changes from other studies: The inference of snow-rich conditions and that such a snow cover insulates the cave cannot be used with much confidence to estimate the full severity of winter temperatures and hence can neither reconstruct nor challenge seasonality changes in a general way.*

There appears to be a misunderstanding about the input parameters the model is considering. The heat-flow model simulates the penetration of the ambient seasonal temperature signal to 50 m depth, and both winter temperature and the temperature buffering effect of a winter snow cover can be varied independently.

The different scenarios are based on regional proxy data reconstructions for the Allerød (Ilyashuk et al., 2009) and the YD (e.g. Affolter et al., 2019; Frauenfelder et al., 2001; Ghadiri et al., 2018; Luetscher et al., 2015). These studies suggest an approximately 3 to 10°C decrease in mean annual air temperatures (MAAT) during the YD compared to modern day. Our experiments take advantage of these studies and investigate under which climate conditions cryogenic cave carbonates (CCCs for short) could have formed at our alpine site (Cioccherloch Cave, Dolomites).

The heat-flow model simulates the penetration of the ambient seasonal temperature signal to 50 m depth. We used local meteorological data to characterize modern day conditions (see Table 2) and palaeotemperature estimates from Majola Pass (Ilyashuk et al., 2009) to define the input parameters for scenario 1 (Allerød interstadial climate). As a recent study by this reviewer (Schenk et al., 2018) suggested that YD summers remained relatively warm with a temperature decreases of 4.3°C in NW Europe and 0.3°C in E Europe relative to the preceding Bølling interstadial, we kept the July temperatures 3-4°C lower modern values (Table 2) and attributed most of the MAAT change to winter cooling. We used the output of this simulation as the starting condition for all early YD experiments. As a second step, we modeled the penetration of the seasonal signal without the presence of winter snow to provide an endmember for the YD cooling (scenarios 2a, 2b, 2d). The results show that the subsurface would be overcooled and prevent CCC formation latest 100 years after the start of the YD.

As a next step we included the buffering effect of a winter snowpack insulating the ground from the winter chill. This buffering effect (snow $\Delta T$) was set to its maximum in scenarios 2c and 2e to test if a similar amplitude of cooling investigated in scenarios 2b and 2d (Table 2) would allow CCC formation given the presence of a winter snow cover. As discussed in the manuscript (lines 304-306), studies of modern permafrost areas suggest that even a 35 cm thick stable winter snow cover may result in a 5.5°C increase in mean ground surface temperature (Zhang, 2005 and references therein). Therefore, our snow $\Delta T$ values of 5°C and 4.7°C are considered realistic. With these two input parameters we characterize the maximum possible amplitude of winter cooling in the absence or presence of a winter snow cover. As the buffering effect of the snow is set to its maximum value to counteract the winter chill, we can indeed reconstruct the possible maximum amplitude of winter cooling with our approach.
In the revised manuscript we will expand the model description and also the description of the experiment.

*In the best case, the results may be valid for the local cave or regional climate setting. However, the authors do not provide evidence for why the results from a cave record at a high elevation from the Southern Alps can generally challenge commonly held views on extreme YD seasonality in other regions and, i.e., not north of the Alps across the Euro-Atlantic region.*

We do not fully understand this comment. By nature every paleoclimate proxy study is local in its significance, be it sediment from a lake, a glacier or a cave. By using proxy data for the YD from the well-studied Alps we compare our winter temperature estimates to other regional proxy data (MAAT and summer temperatures). We emphasize that our data are among the first winter proxy data for the Alps. And we clearly state in the title already that this study concerns a site in this mountain range. Nowhere does it challenge the climate interpretation of the YD in the "Euro-Atlantic region".
Our study provides physical, i.e. non-biological evidence for a maximum amplitude of ≤3°C cooling (in MAAT) at the Allerød-YD transition and argues for a maximum 5.4°C increase in seasonality for the Southern Alps (see Discussion), challenging the notion of extreme seasonality in this region put forward by previous authors working in the Alps (see Discussion). We stand by this interpretation.

*As discussed by the authors, the timing and amount of snow cover over the cave has a major control on subsurface temperature changes. An early and/or thick snow cover will protect the cave from the most severe winter cooling like in January. It is therefore quite likely that the cave record fails to estimate the full winter cooling which would define the amplitude of seasonality change (this might also apply to other studies affected by ice or snow cover insulation). This would imply that the authors do not reconstruct the full seasonality. Hence, they cannot challenge seasonality results from other regions. It is unclear to me how assumptions in the thermal modelling can account for the combination of two unknowns: an unknown winter severity in the YD together with an unknown snow thickness. I think this should be clarified in the text.*

Please see our comment above on capturing the full winter signal by the heat conduction model. In short: because CCC provide a very robust thermal reference point we can place reliable constraints on the amount of winter cooling at this sensitive site. Snow cover modulates the subsurface cooling and we consider this explicitly in the model.
We will attempt to improve the wording in the revised manuscript.

*Line 37: "Siberian-like" would imply extreme seasonality which is typical for continental climates. Such a climate is reasonable for the YD north of the Alps as the major heat source is*

*shut off with an ice-covered North Atlantic Ocean. This does not need to apply to the Southern Alps, though.*

This sentence will be changed according to suggestion made by the colleague and Reviewer#2.

*Line 39: Remove "however" as the sentence before is about winter and this sentence is about summer.*

"however" will be removed.

*Line 39-40: Replace "shutdown" with slowdown - the study assumes an AMOC slowdown of around 36% relative to the Alleröd. The 4.3 to 0.3 summer cooling refers to chironomid-based estimates and hence a notable summer cooling – the mild YD summers are based on plant indicator species and the climate model which suggest no overall summer cooling.*

Thank you for spotting this mistake, will be corrected.

*Line 49: The -10 K change in the northern Alps would be an indication that your record from the Southern Alps does not reflect the same changes – hence you cannot challenge large seasonality changes in general. If anything, your study reflects local or even only cave ambient temperatures/snow cover changes and additional information is required to claim these muted changes would generally apply to south of the Alps. Spagnolo & Ribolini (2019, see section 4.3) estimate a seasonality of 21 degrees for the maritime Alps at the ELA during the YD compared to 14.2 degrees today.*

First, we do not challenge "seasonality changes in general". We compare or results to the Alps. Second, we are aware of the Spagnolo and Ribolini (2019) paper but do not see the suggested offset to our study. Our data suggest a maximum temperature cooling of ≤3°C in MAAT for the YD compared to the Allerød. If we assume a minimum 0.3°C change for July air temperature (see discussion lines 289-302), it is possible to calculate the maximum amplitude of seasonality between July and January air temperatures by using July air temperature and MAAT estimates. Thermal modelling in combination with proxy data of July air temperature shows that seasonality increased by a maximum of 5.4°C at the Allerød-YD transition (see discussion lines 289-302).
Spagnolo and Ribolini (2019) report a winter-summer temperature difference of 21 °C compared to the 14.2°C today, which is very similar to our result. Based on meterological station data the difference between July and January air temperatures at the altitude of Cioccherloch Cave is 17°C.

For the temperature estimates and the calculation of seasonality please see the table below for our site at 2270 m a.s.l.

|                          | Modern | Allerød | YD    |
|--------------------------|--------|---------|-------|
| MAAT (°C)                | 2.5    | 0.5     | -2.5  |
| July air temperature (°C)| 11     | 9       | 8.7   |
| January air temperature (°C) | -6 | -8      | -13.7 |
| Seasonal amplitude (°C)  | 17     | 17      | 22.4  |

We will provide further information to clarify how the value of 5.4°C was calculated. Note that there was a typo in the manuscript (line 301) and the correct value is 5.4°C.

*Lines 62-63: While it is true that Hepp et al. 2019 claim the opposite, several comments by E. Schefuss, B. Zollitschka as well as D. Sachse and myself (see the online discussion: ("https://cp.copernicus.org/articles/15/713/2019/cp-15-713-2019-discussion.html) raise serious questions regarding the validity of that study. The extensively studied Meerfelder Maar nearby does not agree at all with the interpretation using a much more reliable chronology. I think this disagreement needs to be at least mentioned here (e.g. relative to Brauer et al., 1999; Bakke et al., 2009). Line 87: Again, there is no reason why the local cave record should "argue against strong winter cooling during the early YD" at other places and perhaps even not at the site itself.*

We were aware of the discussion on the work by Hepp et al. (2019), however we felt that this study should be mentioned as well in order to provide a full literature review.

*345-350: This might be true for the local setting in the cave insulated by a snow cover but might fail to reconstruct the full MAAT cooling which happened in air temperatures above the snow cover.*

Please see our previous comments.

**References**

Affolter, S., Häuselmann, A., Fleitmann, D., Edwards, R.L., Cheng, H., Leuenberger, M., 2019. Central Europe temperature constrained by speleothem fluid inclusion water isotopes over the past 14,000 years. Sci. Adv. 5, eaav3809. https://doi.org/10.1126/sciadv.aav3809

Frauenfelder, R., Haeberli, W., Hoelzle, M., Maisch, M.A.X., 2001. Using relict rockglaciers in GIS-based modelling to reconstruct Younger Dryas permafrost distribution patterns in the Err-Julier area , Swiss Alps 55, 195–202. https://doi.org/10.1080/00291950152746522

Ghadiri, E., Vogel, N., Brennwald, M.S., Maden, C., Häuselmann, A.D., Fleitmann, D., Cheng, H., Kipfer, R., 2018. Noble gas based temperature reconstruction on a Swiss stalagmite from the last glacial–interglacial transition and its comparison with other climate records. Earth Planet. Sci. Lett. 495, 192–201. https://doi.org/10.1016/j.epsl.2018.05.019

Hepp, J., Wüthrich, L., Bromm, T., Bliedtner, M., Schäfer, I.K., Glaser, B., Rozanski, K., Sirocko, F., Zech, R., Zech, M., 2019. How dry was the Younger Dryas? Evidence from a coupled δ2H and δ18O biomarker paleohygrometer applied to the Gemündener Maar sediments, Western Eifel, Germany. Clim. Past 15, 713–733. https://doi.org/10.5194/cp-15-713-2019

Ilyashuk, B., Gobet, E., Heiri, O., Lotter, A.F., van Leeuwen, J.F.N., van der Knaap, W.O., Ilyashuk, E., Oberli, F., Ammann, B., 2009. Lateglacial environmental and climatic changes at the Maloja Pass, Central Swiss Alps, as recorded by chironomids and pollen. Quat. Sci. Rev. 28, 1340–1353. https://doi.org/10.1016/j.quascirev.2009.01.007

Luetscher, A.M., Hellstrom, J., Müller, W., Barrett, S., 2015. Title : A strong seasonality shift during the Younger Dryas cold spell in the European Alps 1–27.

Schenk, F., Väliranta, M., Muschitiello, F., Tarasov, L., Heikkilä, M., Björck, S., Brandefelt, J., Johansson, A. V., Näslund, J.-O., Wohlfarth, B., 2018. Warm summers during the Younger Dryas cold reversal. Nat. Commun. 9. https://doi.org/10.1038/s41467-018-04071-5

Spagnolo, M., Ribolini, A., 2019. Glacier extent and climate in the Maritime Alps during the Younger Dryas. Palaeogeogr. Palaeoclimatol. Palaeoecol. 536, 109400.

https://doi.org/10.1016/j.palaeo.2019.109400

Zhang, T., 2005. Influence of the seasonal snow cover on the ground thermal regime: an overview. Rev. Geophys. 43, RG4002. https://doi.org/10.1029/2004RG000157

---

## Author Response (AR1)

**Dear Editor,**

We acknowledge your helpful comments and suggestions, and thank the reviewers for their thorough discussion. In the following, we address your comments (in italics) point-by-point.

*With regard to reviewer 2: Please can you explain how figure 6 backs up your conclusions?*
*Please describe the evidence shown in figure 6 and how it evidences your conclusions*

1. Figure 6 – which we updated in the revised version – shows well-established proxy data from Europe (and NGRIP from Greenland) as a framework for temperature and precipitation changes across the Younger Dryas (YD). We selected primarily archives from around the Alps and those that are of high resolution and well dated. We then compare the timing of the CCC occurrences in Cioccherloch cave to these proxy time series.

2. CCC formation at Cioccherloch cave marks stable temperatures very close to 0°C in the early YD at this high-elevation site (which today is at +2.5°C). This tightly constrains the temperature drop at the Allerød-YD transition to ≤ 3°C as there is no proxy evidence suggesting that the Allerød was warmer than today. This value is significantly smaller than the temperature drop suggested by fluid inclusion-based speleothem data from Milandre cave in the Jura Mountains (Affolter et al., 2019; Ghadiri et al., 2018), but is consistent with the moderate YD cooling suggested by paleo-rock-glacier data from the Swiss Alps (Frauenfelder et al., 2001).

3. Autumns and early winters in the early part of the YD were relatively snow-rich, resulting in a stable winter snow cover. This is a robust and novel result of our thermal modelling and is consistent with glacier data showing an early YD maximum extent (e.g. Baroni et al., 2017; Heiri et al., 2014b; Ivy-Ochs et al., 2009; Kerschner and Ivy-Ochs, 2008).

4. In contrast to the Northern European Plain and the Iberian Peninsula (e.g. Baldini et al., 2015, 2019; Bartolomé et al., 2015; Brauer et al., 2002; Lane et al., 2013), high resolution proxy records from the Alps lack evidence of an abrupt climate shift during the mid-YD (Affolter et al., 2019; Ghadiri et al., 2018; Von Grafenstein et al., 2000; Lauterbach et al., 2011; Li et al., 2020; Wurth et al., 2004). CCC formation at this high-alpine cave is consistent with these temperature-sensitive archives and advocates for a small atmospheric warming only (i.e. +1°C in MAAT). In contrast, our data suggest a significant change in precipitation, i.e. a reduction in late season snow fall in the late YD, again consistent with palaeoglacier data.

5. This suggests that the popular model of a south-north migration of the polar front and a concomitant increase in westerly-driven precipitation (e.g., Lane et al., 2013) may be too simplistic at least for the greater Alpine realm, underscoring the need for regionally resolved paleoclimate models.

*With regard to the review of Frederic Schenk: Please can you provide a discussion of why your results from a cave record at a high elevation from the Southern Alps can be used to challenge commonly held views on extreme YD seasonality in other regions?*

Terrestrial proxy records provide strong evidence that the YD climate in central Europe was significantly colder and/or drier and characterized by enhanced seasonality than the Bølling-Allerød (Brauer et al., 2008; Heiri et al., 2014b; Schenk et al., 2018; Affolter et al., 2019; Ghadiri

et al., 2018). Evidence from bio-archives, however, shows that the YD summers were mild and only up to a few °C colder than those in the Bølling-Allerød (Lotter et al., 2000, Ilyashuk et al., 2009; Schenk et al., 2018). The strong drop in mean annual air temperature (MAAT) and mild summers can only be reconciled by a disproportionally strong winter cooling.

Our cave record partly challenges this view of a high seasonality from the perspective of a high-Alpine site, providing tight physical constraints that the MAAT cooling at the Allerød-YD transition there was only ≤ 3°C. We note that our findings are consistent with the only other data available for high elevations in the Alps, palaeoglaciers (Frauenfelder et al., 2001).

Nevertheless, in the light of the criticism of the reviewers and the editor, we modified this relevant sentence in the abstract.

*Please could you clarify what is it about your record, and what evidence is there, that means you can make inferences beyond the local area? Please could you also clarify and provide a fuller explanation for why you can claim that the changes shown in your record apply to the south of the Alps?*

With all the respect, we are a bit puzzled by this remark from Frederik Schenk. By nature, every paleoclimate proxy record is local in its significance, be it a lake, a glacier or a cave. The original manuscript provides a detailed discussion (1) on the link between the cave's microclimate and the ambient climate, (2) on the process leading to CCC formation when cave air temperature is very close to 0°C (Žák et al., 2012, 2018 and references therein), (3) on quantitative climate inferences using a heat flow model (which was never done before in studies of CCC), (4) and on the careful comparison of our record with other proxy records providing insights into YD MAAT, summer temperatures and precipitation.

Although our record is from a single site only, by dating several CCCs of different crystal habitat, we provide a replicated record which is consistent with YD paleotemperature (MAAT) estimates based rock glaciers in the Swiss Alps (Frauenfelder et al., 2001) and a recently published paleoprecipitation reconstruction by Rea et al. (2020).

We intensively discussed the review comments in our group and find no flaws in the data. We acknowledge that the archive we utilize is still poorly known outside the speleothem community and hence there may be some reservations. Fact is that CCCs provide tight and precisely dated physical constraints on the thermal regime of the shallow subsurface on multi-annual time scales and provide the opportunity to draw quantitative inferences about the palaeoclimate, in the case of our cave site about temperature and precipitation. In this respect our study is novel.

Sincerely,

On behalf of all the co-authors,
Gabriella Koltai

Submission of reply to the comments made by Reviewer #1
Ms. Ref. No.: CP-2020-107
Title: Cryogenic cave carbonates in the Dolomites (Northern Italy): insights into Younger Dryas cooling and seasonal precipitation

**Reviewer #1**

We are grateful for the positive and helpful comments and we address below the points raised by this referee (in italics).

Sincerely,

On behalf of the co-authors,
Gabriella Koltai

*Main comments*

*This is an interesting manuscript investigating the internal climatic structure of YD, which in my opinion warrants publication, but I do think further reflection is probably in order. Since much of the paper's discussions/conclusions are drawn from the results of the thermal modeling experiments, it would be helpful if the method is presented in more details (maybe some references would also be appropriate). This could be done by adding additional text in the manuscript or supplemental material section.*

The section of the thermal model has been significantly changed in the revised manuscript and each modelled scenario is now described in the method section.
The thermal model was developed by Alexander H. Jarosch (a colleague who is be a co-author on the revised manuscript). The code is freely available and the relevant references are provided in the manuscript (line 131, lines 365-367).

*The attempt to characterize the early YD climate, basically using only two (at most three) dated CCC may be a little too far reaching, especially given their associated errors. I think authors need to present more convincing evidence for their early YD discussion since three of the scenarios (2b to 2d) are somehow marginally supported by only two CCC samples. This situation is in great contrast with the late YD, for which ~7 dated samples exist. Better explaining why and how scenarios 2b to 2d are really relevant to the discussion would be helpful.*

We would like to emphasize that only the cleanest samples were selected for $^{230}$Th dating to avoid large uncertainties due to high amounts of initial $^{230}$Th. Some of the samples still yielded somewhat larger uncertainties (up to 2.7%). We tried to reduce the error on FOS12-A3 by doing a replicate measurement. Due to the small size of this sample, however, the remainder of this crystal had to be analysed, which resulted in low $^{230}$Th/$^{232}$Th atomic ratios and hence a larger $^{230}$Th age correction. Replicating the $^{230}$Th ages of the other two samples (FOS12-B4 and FOS12-C) was not possible as there was not enough sample left. During the review process, however, we dated two more CCCs from heaps A and C. These analyses yielded $^{230}$Th ages of 12.34±0.2 ka and 12.33±0.2 ka BP, confirms the existing data and are included in the revised manuscript.

During original manuscript preparation we critically evaluated our age data (lines 158-169) and provided discussion on whether CCCs in Cioccherloch Cave represent one prolonged period of CCC deposition lasting for 400-600 years or two distinct phases centred at ~12.6 and 12.2 ka BP. The start of CCC formation in Cioccherloch Cave is at 12.60±0.2 ka and the precision of this [230]Th age is 1.5 %, meaning that CCC deposition commenced earliest at 12.8 ka and latest at 12.4 ka BP (2 sigma uncertainty range). This in combination with the two new ages provides strong evidence that CCC formation indeed started during the early Younger Dryas and conditions allowing CCC formation may have been met for hundreds of years. Such prolonged periods of CCC formation are not unique to Cioccherloch Cave and have been reported form several alpine caves (e.g. Luetscher et al., 2013; Spötl and Cheng, 2014; Spötl et al., in review).

Our model run simulations for the early YD (scenarios 2a-2e) explore the prerequisites for CCC deposition under different climate conditions. We emphasize that all scenarios are based on published regional proxy records providing temperature estimates for the YD cooling (lines 186-208). CCC formation at Cioccherloch Cave can be best explained by scenarios 2c and 2e, arguing for a maximum 4.5 to 5°C drop in MAAT relative to today.

*Authors are using various input parameters for their thermal modeling and end up presenting MAAT,ΔMAAT, MAET,ΔT, snowΔT, etc., point at which tracking all these values in sections 4.4.1 to 4.4.3 is rather difficult and easy to mix up digits. Further-more, they don't always match with what is reported in Table 2 or figures caption (e.g., scenario 2d and 2e is said to be forced with a MAAT of -1.5∘C in the caption of figure3 (line 534), but in Table 2 it appears to be -2∘C; line 535 reads "...ΔT of 4.5∘C" for scenario 2e, but in Table 2 the value is 4.7∘C. The presentation of data in these sections needs to ve revised and made clearer. One way would be to add all values use in Table 2 so that is easier to track them. On the same vein, I see authors derived and reported in Table 2 the mean annual effective temperature, but nowhere in text these values are discussed.*

Thank you for pointing this out. The method section was expanded with the description of each scenario and the input parameters is now consistent throughout the text, figures and Table 2.

*I found Figure 3 to be rather difficult to understand. Some minor improvements, such as placing "early YD" in the right side of the plot and making the blue dashed line more visible, would certainly improve it. However, as expressed above, it is unclear to me which of the CCC really characterize scenarios 2b, c, and d, as I see only two ages with 400 to 600 yrs error that could be assigned to early YD.*

The reviewer probably refers to Figure 5. We followed the suggestions of the reviewer and improved the figure.

*Specific comments*

*I also have many small comments and suggestions that I think would improve the language and clarity of the manuscript .I suggest using throughout the text, figures and captions capital letter C in Cave when is a proper name (e.g., Hölloch Cave, Milandre Cave, etc.).*

Cave names changed.

*Line 60: what do you mean by "certain proxy properties"?*

We refer to the Ti count rate and varve thickness.

*Line 67: Authors could probably make use of the recently published study of Cheng et al. 2020 in PNAS*

Thank you for the suggestion, we are of course aware of this publication (two of the co-authors are the authors of it) but we did not want to cite it before the manuscript was printed. This is included in the revised version.

*Line 77: CCC are in fact speleothems not cave sediments, thus, I urge authors to consider them as such.*

It was corrected.

*Line 84: maybe "CCC form in caves with perennial ice..." will be more clearer to readers than "CCC form within perennial cave ice..."*

We disagree as the suggested change would modify the meaning of the sentence. It is important to emphasize that CCCs form inside the cave ice bodies.

*Line 105: Methods - is there any other way of presenting the information in this chapter without breaking it so heavily and have only 2-3 lines for various subchapters?*

We looked at the manuscript guidelines and also at other manuscripts published in this journal and think that breaking the methods down into subsections is still the best way to present them.

*Line 129: Thermal modeling - additional information and references are needed in order to better understand the method (e.g., what might be the effect of taking 0.5 or 1for dT/dz? What are the uncertainties of the results associated with this model?)*

*and*

*Line 131: a reference to whoever generated the heat flow model would be appropriate*
*Line 135 Equation 1 - if authors consider dT/dz = 0 then Q is 0 regardless of thermal conductivity, right? Do I miss something on how this equation really helps?*

*and*

*Line 136: thermal diffusivity (how fast heat diffuses through a material) is not the same with thermal conductivity (ability of a system to transport heat energy). Authors define thermal conductivity as "c" in Equation 1, but then set the thermal diffusivity of lime-stone to 1.2x1010^-6. What value was actually used for Eq. 1, which once again, ifdT/dz is assumed 0, Q would be 0*

About references on the model, please see our comment above. The model description section was modified to clarify these misunderstandings. The model solves the basic heat equation

$$\frac{\partial T}{\partial t} = \alpha \frac{\partial^2 T}{\partial z^2}$$

where here α is the thermal diffusivity and T temperature and z the vertical coordinate. This is a standard model approach well known and understood. The numerical errors are minimal. Equation (1) in the manuscript describes the relation between heat flux and thermal gradient. This equation, however, is not solved in the model. The thermal gradient (dt/dz) was set to zero as a lower boundary condition of the model.

As mentioned above the method section was expanded to avoid any misunderstanding.

*Line 150 - 153: For consistency, use XX‰ or XX ‰ but not both ways.*

The space in line 150 was deleted.

*Lines 180-223: I feel that the presentation of these scenarios could be clearer if all values are included in Table 2 or those already in this table are presented in text as well. Right now, I see, for some scenarios, different values in text, table, and/or caption of figures 3 and 4.*

This is improved.

*Lines 274-275: what do you mean by "Scenario 2e including provides...*

"Scenario 2e including a winter snow cover provides…". Thank you for pointing it out.

*Line 301: it was not immediately clear to this reviewer how the value of 5.7◦C was derived. Please add text to clarify.*

Thank you for spotting this mistake. There was a typo in the sentence and in fact this value is 5.4°C. Text was added to clarify how this value (5.4°C) was derived.

*Lines 346-347: my suggestion for rewording this part of the sentence: "...CCC in the Dolomites, which in contrast to many studies from Central European caves, formed not during..."*

Following the reviewer´s suggestion we changed this in the manuscript.

*Line 358: add "for" at "advocates for a mild..."*

Thank you for the suggestion.

*Line 542 - Figure 4 - winter snow cover (ΔT = 2◦C) is mentioned in this caption, but it is not in Table 2 for scenario 3a*

This was corrected.

Submission of reply to the comments made by Reviewer #2
Ms. Ref. No.: CP-2020-107
Title: Cryogenic cave carbonates in the Dolomites (Northern Italy): insights into Younger Dryas cooling and seasonal precipitation

**Reviewer #2**

We thank the reviewer for his/her critical comments. We address below the points raised by this referee (in italics) and try to clarify what did in response to these comments (blue).

Sincerely,

On behalf of all the co-authors,
Gabriella Koltai

*Main comments*

*Koltai et al. use a combination of cave air temperature modeling and U/Th dating of cryogenic cave carbonates (CCC) to discuss climate variability during the YD in the SE Alps (Dolomites). The topic is becoming a hotly debated one, as more and more studies suggest that climate during the YD was spatially, temporally and seasonally different throughout Europe and the author's paper comes to add more information to this debate. While bot the title and discussion are tantalizing, I fond that the authors overstretch themselves in analyzing to many climate variables over a long period of time based on a limited data set (several U/Th dates) buttressed by modeling. Several points should be made clear before publication of the paper. I detail these in a few general and technical comments below.*

The description of our approach may have been unclear and we are thankful for pointing this out. We have made major changes in the methods section of the revised manuscript. This paper utilizes a novel paleoclimate archive (cryogenic cave carbonates, or CCCs for short) that allows to precisely constrain permafrost thawing events in the past, when the cave air temperature was very close to the freezing point (e.g. Zak et al., 2018 and references therein, for further discussion on CCC formation see lines 76-85 and 225-266).

In our study we use a 1-d heat conduction model developed by one of the co-authors to investigate how the atmospheric climate signal is transferred into the subsurface. The different scenarios are based on regional proxy data reconstructions for the Allerød (Ilyashuk et al., 2009a) and the YD (e.g. Affolter et al., 2019; Frauenfelder et al., 2001; Ghadiri et al., 2018; Luetscher et al., 2015). These studies suggest an approximately 3 to 10°C decrease in mean annual air temperature (MAAT) during the YD compared to modern day. Our experiments take advantage of these studies and investigate under which climate conditions CCCs could have formed at our study site.
The heat-flow model simulates the penetration of the ambient seasonal temperature signal to 50 m depth. We use local meteorological data to characterize modern day conditions (see Table 2) and palaeotemperature estimates from the Majola Pass (Ilyashuk et al., 2009b) to define the input parameters for scenario 1 (Allerød interstadial climate). As a recent study by Schenk et al. (2018) suggested that YD summers remained relatively warm with temperature decreases of 4.3°C in NW

Europe and 0.3°C in E Europe relative to the preceding Bølling interstadial, we kept the July temperatures 3-4°C lower modern values (Table 2) and attributed most of the MAAT change to winter cooling. We use the output of this simulation as the starting condition for all early YD experiments. As a second step, we model the penetration of the seasonal signal without the presence of winter snow to provide an endmember for the YD cooling (scenarios 2a, 2b, 2d). The results show that the subsurface would be overcooled and prevent CCC formation latest 100 years after the start of the cooling. Then as a next step we include the buffering effect of a winter snowpack insulating the ground from the winter chill. This buffering effect (snow ΔT) is set to its maximum in scenarios 2c and 2e to test if a similar amplitude of cooling investigated in scenarios 2b and 2d (Table 2) would allow CCC formation given the presence of a winter snow cover. As discussed in the manuscript (lines 304-306) studies in modern permafrost areas suggest that even a 35 cm thick stable winter snow cover may result in a 5.5°C increase in mean ground surface temperatures (Zhang, 2005 and references therein). Therefore, our snow ΔT values of 5°C and 4.7°C are considered to be realistic for the YD at this alpine setting. With these two input parameters we characterize the maximum possible amplitude of winter cooling in the absence or presence of a winter snow cover.

*Several papers discussed climatic inferences based on the U/Th age of CCCs and a wide range growth periods have been found leading to several possible climatic conditions leading to the precipitation of CCC (Zak et al., 2004, 2009, 2012, Luetscher et al., 2013, Spoetl and Cheng, 2014 – quite a few of these are missing from the cited literature section. . .). These authors have found CCS growing during warm and cold, dry and wet periods during MIS6, MIS4, MIS3, MIS 2, mid-to-late Holocene (Roman and Medieval Warm Periods). From these studies, it occurs that a wide range of external climatic conditions are possibly favourable for the formation of CCCs in caves and it is the peculiarities of cave climate that are in the end responsible for this. Consequently, I find the climatic inferences made in this paper somehow only poorly supported by the data but strongly relying on the thermal modeling. While the data are what they are, the modeling methods and results should be explained in more detail and the various assumptions (e.g., lack or presence of permafrost, assumed temperatures, buffering effect of snow cover etc) in choosing input data and favouring one model over the other better explained.*

We agree with the reviewer that the local cave microclimate may influence CCC formation to a considerable extent in complex cave systems (cf. also Koltai et al., 2020), however we are confident that this was not the case during the YD in Cioccheroch Cave. In the manuscript (lines 96-101) we present the temperature data of a 1-year monitoring of the CCC-bearing cave chamber and provide a discussion the microclimate of the site (lines 238-253).

Regarding the modeling methods, we refer to our previous comment from Reviewer#2. We would like to emphasize that all previous studies interpreted CCCs as proxies for paleo-permafrost thawing. In our paper we go a step further and apply a 1-d heat flow model to characterize under which climate conditions CCCs could have formed in the cave. We emphasize in the manuscript that CCCs form under a stable cave microclimate when the cave air temperature is negative and very close to the 0°C isotherm. If the cave would have been strongly ventilated and heat advection played an important role, the fine crystalline variety of CCCs (essentially crystal powders) would have formed instead of the coarse crystalline one. The innovative aspect of our study is the quantitative link between the climate signal recorded by the CCCs to the surface environment.

The references were added to the manuscript.

*Further, the authors could summarize the climatic conditions during the YD in a simplified figure, emphasizing the seasonally distinct climatic conditions and the two-part YD climate and than add their data in support of the inferred climatic conditions. The concluding figure 6 does not clearly supports the authors' claims.*

6. Figure 6 – which we updated in the revised version following the reviewer´s suggestion – shows well-established proxy data from Europe (and NGRIP from Greenland) as a framework for temperature and precipitation changes across the Younger Dryas (YD). We selected primarily archives from around the Alps and those that are of high resolution and well dated. We then compare the timing of the CCC occurrences in Cioccherloch cave to these proxy time series.

7. CCC formation at Cioccherloch cave marks stable temperatures very close to 0°C in the early YD at this high-elevation site (which today is at +2.5°C). This tightly constrains the temperature drop at the Allerød-YD transition to ≤ 3°C as there is no proxy evidence suggesting that the Allerød was warmer than today. This value is significantly smaller than the temperature drop suggested by fluid inclusion-based speleothem data from Milandre cave in the Jura Mountains (Affolter et al., 2019; Ghadiri et al., 2018), but is consistent with the moderate YD cooling suggested by paleo-rock-glacier data from the Swiss Alps (Frauenfelder et al., 2001).

8. Autumns and early winters in the early part of the YD were relatively snow-rich, resulting in a stable winter snow cover. This is a robust and novel result of our thermal modelling and is consistent with glacier data showing an early YD maximum extent (e.g. Baroni et al., 2017; Heiri et al., 2014b; Ivy-Ochs et al., 2009; Kerschner and Ivy-Ochs, 2008).

9. In contrast to the Northern European Plain and the Iberian Peninsula (e.g. Baldini et al., 2015, 2019; Bartolomé et al., 2015; Brauer et al., 2002; Lane et al., 2013), high resolution proxy records from the Alps lack evidence of an abrupt climate shift during the mid-YD (Affolter et al., 2019; Ghadiri et al., 2018; Von Grafenstein et al., 2000; Lauterbach et al., 2011; Li et al., 2020; Wurth et al., 2004). CCC formation at this high-alpine cave is consistent with these temperature-sensitive archives and advocates for a small atmospheric warming only (i.e. +1°C in MAAT). In contrast, our data suggest a significant change in precipitation, i.e. a reduction in late season snow fall in the late YD, again consistent with palaeoglacier data.

10. This suggests that the popular model of a south-north migration of the polar front and a concomitant increase in westerly-driven precipitation (e.g., Lane et al., 2013) may be too simplistic at least for the greater Alpine realm, underscoring the need for regionally resolved paleoclimate models.

*Specific comments*

*28 – GS1 starts at 12.9 ka, not 12.8 ka (Rasmussen et al., 2014)*

The start of GS1 was at 12.896 yr b2k (Rasmussen et al., 2014), which is 12.846 yr BP.

*35 – catastrophic is rather human-centered*

We changed this.

*37 – perhaps "cold" is enough, Siberian-like is quite subjective (and given that this is a paleoclimate paper, Siberian climate varied widely in the past)*

We rephrased this.

*41 – relative to. . .?*

Relative to the Bølling-Allerød interstadial (see line 41)

*48-52 – to which season do these reconstructions refer?*

These reconstructions refer to annual air temperature as stated in line 50.

*69-70 – not clear how "enhanced precipitation differences between the northern, central and southern part of the Alps" would result in a YD maximum. Also, check the 13.5 ka age, it is well before the onset of the YD*

We rephrased this sentence for clarification and added a new reference added for the 13.5 ka BP age (Ivy-Ochs, 2015).

*73 – what do you mean by "double response"? Two periods of glacier advance? Please clarify*

Yes, we mean two glacier advances as reported by Baroni et al. (2017).

*86-88 – I particularly enjoy this statement, but please clarify 1) what do you mean by "strong winter", 2) what season the "1-2 C warming" refers to and 3) the reference for the 'drier' comparative (e.g., "drier" compared to early YD?)*

This is rephrased in the revised manuscript.
For clarification
   (1) by strong winter cooling we refer to the 10°C decrease in MAAT compared to present day, as reported from the Jura Mountains (Affolter et al., 2019; Ghadiri et al., 2018), which requires a disproportionally large winter cooling considering that the summer cooling for this part of Europe was rather small as shown by pollen and chironomid data (as detailed in the ms.).
   (2) the 1-2°C warming refers to the MAAT
   (3) and drier is meant relative to the early YD (as stated in line 89)

*91 – This sentence is a odds with the cave's description here*

We removed this sentence.

*103-104 – over what period were these snow depth values measured? Snowfall heights do not record the amount if snowfall accurately, please provide the total amount of winter and early winter (September-December) precipitation.*

Snow height is monitored in the Dolomites at Rossalm and Piz la Ila stations. Unfortunately, the total amount of precipitation is not measured. We used the monthly data for the last seven years (2012-2019) from Rossalm and for a fifteen-year-observation-period (1999-2014) at Piz la Ila to calculate average snowfall amounts for the autumn (September to December).

*114 – What was exactly sampled for stale isotope analyses? Entire CCC? Outer/inner part f it? Please detail.*

Most of the CCCs were too tiny to be cut therefore a handheld drill was used to take small aliquots of carbonate powder for stable isotope analyses. Usually, the outer layer was drilled off and discarded and then the carbonate powder was drilled 1-2 mm below the surface. In case of the dated CCC samples, a small aliquot of the drilled carbonate powder was used for stable isotope analyses.

*120 – I understand that these CCCs grow over prolonged periods of time. What part of the individual CCCs was sampled for dating? Or was it whole sample?*

This is partially explained in the manuscript (line 120-122), and more clarified it in the revised manuscript. If the single crystals were large enough, the carbonate powder was drilled from the center of the CCC to define the start of CCC formation (15 samples). In case of two skeletal crystals, the entire crystal was used for $^{230}$Th dating.

*200 – why was a 5 _C temperature chosen for the buffering effect of snow cover?*

Please see our response above.

*230 – how long does it take for these CCCs to form? Several years is not that much in terms of YD climate variability, so with only 3 ages for the early YD, the inferences made in this article might be slightly far-fetched*

The reviewer asks a long-standing question in the community working on CCCs. The short answer is that we only know that the fine crystalline variety of CCCs (essentially crystal powder) forms within a matter of hours to days by comparably rapid freezing of a water film. The coarse crystalline CCC variety – the one we talk about in this study – has never been observed in statu nascendi, and nobody has made experiments growing them under controlled conditions. Still, there is unanimous consensus among colleagues working on coarse crystalline CCC that these carbonates form (a) not within a water film but in freezing pools in the ice, and (b) require much longer to form than their fine crystalline counterparts. This is pretty obvious given the well-developed macroscopic crystals and the fact that in other caves CCC can reach several cm in diameter. There is no published information how much time is involved in the formation of individual coarse crystalline CCC aggregates. In our group we also study Pleistocene CCCs in Siberian caves and they can be up to a few cm in diameter. We dated core and rim of these large CCCs. Even the most precise $^{230}$Th ages cannot resolve an age difference within individual CCCs. Considering the age uncertainties up to several hundred years could be involved in the growth history of some of these aggregates. Although the Cioccherloch samples are significantly smaller than their counterparts from Siberia

we are convinced that months to years (and decades in the case of larger particles) are likely involved in their formation. These assumptions are also supported by the smooth isotope and trace element distribution patterns of these particles.

We disagree with the reviewer on the inferences being too far-fetched. During the review process we dated two more CCCs from heaps A and C. These analyses yielded $^{230}$Th ages of 12.34±0.2 ka and 12.33±0.2 ka BP, providing further support for CCC formation during the early YD. We strongly believe that neglecting the possibility of early YD CCC formation (as supported by the $^{230}$Th ages and their 2σ uncertainties) would be an oversimplification.

*235-236 – here is a bit of a jump in logic, as a few lines above (230) a few years are required for CCC to form and now the suggestion is that cave climate was stable for centuries.*

We do not see a problem here, and please see our comment above on CCC formation. The majority of $^{230}$Th ages overlap within their 2σ errors. Similar prolonged periods of CCC formation have been reported form other alpine caves (e.g. Luetscher et al., 2013; Spötl and Cheng, 2014, Spötl et al., in review)

*245 – do you have any indication on when ruble closed the connecting gallery? It could have been open during the YD and hence the cave would have been out of thermal equilibrium with the outside, as discussed above*

We have no information on when the rubble closed the narrow connection.

*246-248 – I would argue that CCC record changes in the thermal state of the cave, that could be or not in equilibrium with external conditions. The assumption is that the ruble blocking the cave was there throughout and since the YD*

We disagree. The presence of a snow cone slightly larger than today could have also sealed this connection. Also, the connection between the CCC-bearing chamber and the entrance shaft could not have been more open in the past than it is today. We provided a discussion on advective processes in the original manuscript in lines 283-298.
We expanded this section in the revised manuscript.

*250-251 – Liquid water reaching the cave would have a dramatic impact on temperature, given the extremely high specific heat capacity of water. I think this is to easily dismissed. And if I understood right, liquid water was required to form CCC (line 259 in the text)*

We considered this process qualitatively and concluded that drip water obviously entered this cave chamber forming meltwater pools on the ice, eventually giving rise to CCC formation. This very slow flowing seepage water, however, is likely thermally equilibrated with the ca. 50 m-thick rock above the cave and given its very low discharge carries comparably little heat from the surface. In addition, the YD climate was likely drier than today (as discussed in the ms.) hence discharge was even lower.
We expanded this section in the revised manuscript.

*264 – U/Th ages show that CCC formed well (several hundreds of years) into the YD, not at the transition.*

This sentence was rewritten.

*274 – how likely is that the cave was perennially frozen throughout both winters and summers for several centuries? If not, than the proposed shielding by snow is not required to induce changes in cave air temperature around 0 and thus facilitate the precipitation of CCC*

CCC only form very close to 0°C and our data therefore rule out that this cave was well below 0°C during the YD.

*277 and subsequent discussion – winter or summer very cold YD? The distinction was heavily promoted in the introduction, it should be made here, as well.*

A separate subsection is devoted to the discussion of seasonality changes (5.3 Increased seasonality in the early YD, lines 289-313)

*289 – see my comment on the ruble blocking the cave and its effect on cave climate. How was the 3 C temperature obtained?*

This maximum amplitude cooling (≤3°C) at the Allerød-YD transition was derived from thermal modeling (scenarios 2c and 2e, please see lines 272-276).

*292 and subsequent –warm YD summers indicate that the cave would have been warm enough to lead to ice melting and prevent the all-over freezing of cave. Consequently, CCC could have precipitated by the freezing of water formed during warm summers on the surface of (cold winter forming) cave ice. Is this a likely scenario?*

That is an interesting suggestion. Warm summers are accounted for in the modeling experiments (Scenarios 2a-2e). However, we the seasonal temperature signal is cancelled out after the first 10 meters of rock (see the figures showing the heat-flow model results).

CCCs cannot form by repeated thaw-freeze cycles of cave ice as there would not be enough dissolved ions in the water to reach supersaturation via freezing and to precipitate cryogenic minerals. Drip water derive from the epikarst and the vadose zone is needed to deliver the solutes necessary for CCC formation. As slow freezing proceeds, the expulsion of these ions leads to supersaturation and consequently the precipitation of CCCs (e.g. Žák et al., 2012, 2008). This is corroborated by U concentrations much higher than those of warm-climate, non-cryogenic speleothems from the same cave (see Table 2 in the revised ms.).

*303-305 – this shielding would not be required if the scenarios presented above could be happening.*

Please see our comment above.

*320 – U/Th ages indicate that CCC formed for several centuries, so why the emphasize on this correlation with the mid-YD transition? The age errors and the widespread ages of all CCCs are to large compared to the narrow age of the transition to support the subsequent discussion.*

The majority of the $^{230}$Th ages cluster at 12.2 ka BP. The weighted mean of all ages is 12.19±0.6 ka BP. As this coincides with the mid-YD transition (12.24 ±0.4 ka BP, see Lane et al., 2013) CCC formation in Cioccherloch Cave was likely connected to this climate event.

[revised manuscript text omitted]

---

## Author Response (AR2)

Dear Editor,

We appreciate the careful reading by Reviewer 1 and accepted many of his suggestions. We also went through the manuscript carefully and made minor corrections in the text, in Figures 1, 5 and 6, in Table 2, in Supplementary Figure 1. and in the list of references. Below we address the point-by-point changes suggested by Reviewer 1.

Lines 16 and 17
We rewrote this part of the abstract.

Line 18
This mountain range is called "the Dolomites". No change made.

Line 67
"The" was replaced by "an".

Lines 135-136
The text was partly changed following the reviewer´s suggestion.
We moved the bottom boundary of the model deeper than the cave to avoid numerical influences of the boundary with the solution in the cave (at cave depth). This is numerically a more sound approach than modelling just to the cave depth. Therefore, our simulations are run to 70 m depths.

Line 145
A reference for the relevant figures was added to the text (Suppl. Figs. 2-4).

Line 148
The typo was corrected.

Lines 173-174
The sentence was moved to the end of the paragraph, as suggested by the reviewer.

Line 183
"Instantaneous" was deleted to avoid confusion.

Line 231
"Scenario" was added to all relevant places in chapters 4.4.2 and 4.4.3

Lines 266-273
We modified these sentences.

Line 288
"During" was replaced by "In".

Line 292
"Cave" was deleted.

Line 311
"To" was added.

Lines 313-314
The sentence was changed.

Lines 360 and 361
Typo was corrected and the repeated word was deleted.

Figure 5
Thank you for spotting this typo in Table 2 (FOS12-C2), it is corrected in the revised manuscript. We double-checked the data and they are correct.

Sincerely,
On the behalf of all co-authors,

Gabriella Koltai